



# The role of land-surface interactions for surface climate in the EC-Earth3 earth system model

Wilhelm May[1]

[1]Centre for Environmental and Climate Science, Lund University, Lund, SE-223 62, Sweden

*Correspondence to*: Wilhelm May (wilhelm.may@cec.lu.se)

**Abstract.** Land-surface conditions have prominent effects on local and regional climate through the exchanges of energy and moisture with the atmosphere and on global climate by the exchanges of carbon dioxide. Therefore, it is important that land-surface components of earth system models (ESMs) like EC-Earth3 can simulate the processes governing the energy and water cycles and the carbon cycle realistically. The aim of this study is twofold, first to evaluate the quality of the simulation of

surface climate by the land-surface component of the EC-Earth3 ESM, combining the HTESSEL land-surface model and the LPJ-GUESS dynamic vegetation model, and second to assess the role of the coupling of the land surface with the atmosphere for the simulation of the surface climate in EC-Earth3. To this end, two simulations with different configurations of the EC-Earth3 ESM are considered: an offline simulation with HTESSEL+LPJ-GUESS forced with meteorological data from the ERA5 re-analyses and a simulation with the atmospheric component of EC-Earth3, where the land-surface conditions (soil

moisture and vegetation characteristics) are prescribed from the offline simulation. The land-surface component of EC-Earth3 is characterized by marked regional biases in various aspects of surface climate. These are, for instance, too warm land-surface temperatures in the tropics and in the mid- and high latitudes of the Northern Hemisphere, resulting in a warm overall bias. Surface soil moisture, on the other hand, is characterized by a dry bias in the subtropics and parts of the extra-tropics and a wet bias in the tropics and the eastern part of Asia, resulting in a slightly negative overall bias. The incoming net radiation is

underestimated by the model over much of the global land area, causing a negative overall bias. For the fluxes of sensible heat, the model also shows a negative overall bias with a clear tendency to underestimate the sensible heat fluxes in regions, where they are relatively strong, and underestimate them in regions where they are rather weak. The biases in the fluxes of latent heat generally correspond to the biases in the sensible heat fluxes (with opposite sign) with an underestimation of the fluxes of latent heat in regions where the sensible heat fluxes are too strong and an overestimation in the regions where the sensible heat

fluxes are too weak. The coupling with the atmosphere leads to somewhat stronger biases in the aspects of surface climate considered in the study. The most pronounced effect of the coupling is found for land-surface temperature, including a change in the sign of the overall bias from a warm overall bias in the simulation with the land-surface component to a considerable cold bias in the atmospheric component of EC-Earth3. For surface soil moisture, the coupling with the atmosphere changes a dry overall bias of the land-surface component to a wet bias in the atmospheric model. Analysing the correspondence between

the global patterns for the simulations and the reference data reveals relatively large effects of the atmospheric coupling on land-surface temperature as well as on net radiation and sensible heat flux but small effects on surface soil moisture and latent heat flux.

## 1 Introduction

Earth system models (ESMs) are an essential tool for simulating climate in general and for simulating future changes in climate

under assumed changes in the emissions of greenhouse gases (GHGs) or changes in land-use and land-cover due to human activities in particular (e.g., O'Neill et al., 2016). ESMs are designed to incorporate all aspects that affect the Earth system. They are capable to explicitly represent biogeochemical processes that interact with the physical climate and, thus, affect its response to forcing by anthropogenic GHG emissions (e.g., Flato et al., 2011). Representing the global carbon cycle, for


instance, enables feedbacks between the physical climate and the biological and chemical processes in the ocean and on land

that take up some of the emitted carbon dioxide ($CO_2$) and, thus, reduce the climate warming.

In its recent version 3 (Döscher et al., 2022), EC-Earth is one of the ESMs that have been extensively used in the Coupled Model Intercomparison Project Phase 6 (CMIP6; Eyring et al. 2016) and have contributed to the 6th Assessment Report of the Intergovernmental Panel on Climate Change with scenario simulations (e.g., Lee et al., 2021). EC-Earth3 can be used in a variety of model configurations, reflecting the wide range of research interests in the EC-Earth community (Döscher et al.,

2022). According to Döscher et al. (2022) , key performance metrics illustrate physical behaviour and biases well in the range of recent CMIP models for EC-Earth3. With improved physical and dynamic features, additional model components representing the Earth system, and largely improved quality of the model, EC-Earth3 represents a distinct step forward from the version of EC-Earth used for CMIP5. Biases of EC-Earth3 include, for instance, an overall cold bias of near-surface temperatures over land as well as a wet bias in precipitation over the land areas in the tropics and the extra-tropics and a dry

bias over the land areas in the subtropics.

Through interactions with the atmosphere, the land surface plays an important role for the state of the atmosphere and, thus, the local and regional climate. These interactions affect the energy and water cycles through the exchanges of energy and water as well as the composition of the atmosphere through exchanges of carbon (e.g., Bonan, 2008) and the release of aerosols and biogenic volatile organic compounds (BVOCs; e.g., Bonan, 2016). As for the state of the land surface, for instance, soil

moisture, has marked effects on climate through couplings with air temperature and precipitation (e.g., Seneviratne et al., 2010). For the coupling with temperature, a negative soil moisture anomaly causes decreasing evapotranspiration. This leads to an increased sensible heat flux and, hence, to warmer near-surface air temperatures. Warmer temperatures, in turn, lead to a higher water vapour deficit and an enhanced evaporative demand, causing a further decrease in soil moisture due to a potential increase in evapotranspiration. A positive soil moisture anomaly, on the other hand, leads to cooler temperatures in response

to increasing evapotranspiration and, hence, a decreased sensible heat flux. The near-surface cooling, in turn, stabilizes the atmospheric boundary layer and, therefore, might, suppress convective precipitation, counteracting the initial positive soil moisture anomaly. For the coupling with precipitation, a negative soil moisture anomaly generally leads to less evapotranspiration and a positive soil moisture anomaly to more evapotranspiration and, thus, to stronger and weaker fluxes of latent heat, respectively. The impact of the induced changes in latent heat flux on precipitation is somewhat uncertain, as

several complex processes are involved. This uncertainty is related to both the direction of the coupling and its strength. Typically, however, increasing precipitation leads to higher soil moisture and decreasing precipitation to lower soil moisture.

The state of vegetation affects climate through biophysical and biogeochemical processes (Bonan, 2016). In the case of changing vegetation by cutting forests, the associated changes in surface albedo, the roughness length and evapotranspiration have distinct effects on climate. The increase in surface albedo when changing from trees to grass reduces the fraction of the

incoming solar radiation that is absorbed at the land surface and, thus, leads to weaker fluxes of sensible heat into the atmosphere. The decrease in the roughness length, on the other hand, weakens the turbulent fluxes of energy and moisture into the atmosphere and reduces mixing in the atmospheric boundary layer. The decrease of evapotranspiration decreases latent heat fluxes and increases the fluxes of sensible heat, resulting in warmer and dryer local climate conditions. Changing from trees to grass also  increases the atmospheric concentration of $CO_2$ and reduces the release of BVOCs leading to a lower

number of aerosols, which affects the radiative balance, the formation of clouds as well as the occurrence and character of precipitation.

Perugini et al. (2017) provided a synthesis of the biophysical effects of anthropogenic changes in land use, i.e., deforestation vs. forestation at regional and global scale, on air temperature and precipitation based on numerous scientific publications considering observations as well as model simulations. As a response to regional deforestation, most simulations

are characterized by a cooling effect in the boreal regions, a slight cooling in the temperate zone and a warming in the tropics. Observations confirm the simulated cooling effect in the boreal regions and the warming in the tropics but show a slight





warming in the temperate zone. Regional afforestation shows the opposite behaviour, with a warming in the boreal regions and the temperate zone and a cooling in the tropics in the model simulations (again observations show the opposite effect in the temperate zone). According to the synthesis, most simulations are characterized by a decrease in annual mean precipitation

in response to regional deforestation in all three climate zones, i.e., the boreal, temperate and tropical zone.

Given the important role of the land surface for local and regional climate it is essential to ensure that the land-surface component of an ESM can simulate the processes governing the energy and water cycles, including the exchanges of energy ad moisture with the atmosphere, as well as the carbon cycle with the exchanges of $CO_2$ with the atmosphere realistically. As part of the LS3MIP intercomparison project, for instance, it has been envisaged to assess the quality of the land-surface

components in ESMs based on offline simulations of land-surface components forced with different meteorological data sets (Van den Hurk et al., 2016). In LS3MIP the focus is on the physical aspects of the land surface considering the representations of surface fluxes, snow cover and soil moisture.

The aim of this study is twofold, first to evaluate the quality of the simulation of surface climate by the land-surface component of the EC-Earth3 ESM, combining the HTESSEL land-surface model and the LPJ-GUESS dynamic vegetation

model (see references below), and second to assess the role of the coupling of the land surface with the atmosphere for the simulation of the surface climate in EC-Earth3. To this end, two simulations with different configurations of the EC-Earth3 ESM will be considered: an offline simulation with the land-surface component of EC-Earth3, HTESSEL+LPJ-GUESS, forced with meteorological data from the ERA5 re-analyses (Hersbach et al., 2020) and a simulation with the atmospheric version of EC-Earth3, where the land-surface conditions, i.e., soil moisture and vegetation characteristics, are prescribed from the offline

simulation. By comparing the offline simulation with observational data, the study addresses the research question of "To which extent is the land-surface component of EC-Earth3, HTESSEL+LPJ-GUESS, capable to simulate surface climate under 'perfect' climate conditions?" in relation to the first aim. And by comparing the simulation with the atmospheric version of EC-Earth3 with the offline simulation (and with the observational data), the study addresses the research question of "To which extent do land-surface atmosphere interactions affect the biases of the surface climate in EC-Earth3?" in relation to the second

aim.

The different model components and configurations of the EC-Earth ESM and the simulations considered in this study are described in Section 2 and the observational data are presented in Section 3. In Section 4, the quality of the simulation of the surface climate by the land-surface component HTESSEL+LPJ-GUESS is evaluated. The role of the coupling of the land surface with the atmosphere for the simulation of the surface climate in EC-Earth3 is assessed in Section 5. A discussion as

well as a summary and conclusions are given in Section 6 and Section 7, respectively.

## 2 Models and simulations

In this study, version 3 of the EC-Earth ESM (Döscher et al., 2022) was applied in different configurations. EC-Earth3 incorporates various modules that describe the atmosphere, ocean and sea ice, the land surface and dynamic vegetation, the atmospheric composition, and the Greenland ice sheet. The atmospheric component is based on cycle 36r4 of the Integrated

Forecast System (IFS) of the European Centre for Medium-Range Weather Forecasts (ECMWF), including the HTESSEL land-surface model. Dynamic vegetation and terrestrial biogeochemistry are simulated by the LPJ-GUESS terrestrial ecosystem model. NEMO3.6 is the ocean component and LIM3 is the sea-ice component of EC-Earth3, and biogeochemical processes in the ocean are simulated by the PISCES model. The TM5 model is used to incorporate aerosols and chemical process in the atmosphere. Finally, the PISM ice sheet model can be included to simulate the Greenland ice sheet. Most of the

different model components are coupled by means of the OASIS-MCT coupling library. Details on the EC-Earth3 ESM as well as on the different model components and various references are given in Döscher et al. (2022). In the specific version described in that paper, EC-Earth3 has extensively contributed to CMIP6 (Eyring et al. 2016).



### 2.1 Land-surface component

The land-surface component of EC-Earth3 includes the HTESSEL land-surface model, which is integrated into IFS, and the
LPJ-GUESS terrestrial ecosystem model. The land-surface component can be run offline, with the climate forcing prescribed
from reanalyses or from simulations with EC-Earth.

### 2.1.1 HTESSEL

The Hydrology Tiled ECMWF Scheme of Surface Exchanges over Land (HTESSEL; Balsamo et al. 2009) is used to describe
the energy and water balances in the ground and at the land surface. This model is an extension of the Tiled ECMWF Scheme
for Surface Exchanges over Land (TESSEL; van den Hurk et al., 2000), correcting for the absence of surface runoff and a
globally uniform soil texture in the preceding model. In addition to the improved soil hydrology, HTESSEL includes a new
snow scheme (Dutra et al., 2010), a multiyear satellite-based vegetation climatology (Boussetta et al., 2013) and a revised bare
soil evaporation (Albergel et al., 2012). Vegetation types and vegetation cover, distinguishing between high and low
vegetation, are either prescribed from a static land use map (Boussetta et al., 2013), interactively coupled with LPJ-GUESS or
prescribed from an existing simulation with LPJ-GUESS. High vegetation comprises various types of trees and low vegetation
includes low-growing vegetation types such as crops, grasses and shrubs. The variables characterising vegetation determine
surface albedo, surface roughness and soil water exploitable by roots (Döscher et al., 2022), strongly affecting the exchanges
of energy between the land surface and the atmosphere. HTESSEL has up to six different land-surface tiles with separate
energy and water balances including bare soil, low and high vegetation, intercepted water as well as snow shaded by high
vegetation and snow on low vegetation, and four soil layers, i.e., 0-7, 7-28, 28-100 and 100-255 cm.

### 2.1.2 LPJ-GUESS

The LPJ-GUESS terrestrial ecosystem model (Smith et al., 2001; 2014) is a second-generation dynamic global vegetation
model (DGVM), in which the dynamics of ecosystems arise from neighborhood-scale interactions between cohorts
(characterized by age and size) of different plant functional types (PFTs; classifications of plants according to their physical,
phylogenetic and phenological characteristics). LPJ-GUESS simulates several replicate patches in each model grid cell,
thereby capturing the variation in a landscape resulting from different trajectories of population dynamics (i.e., establishment
and mortality) and disturbance. As part of EC-Earth3, LPJ-GUESS distinguishes between 11 PFTs, i.e., three types of boreal
trees (boreal needleleaved evergreen, boreal shade-intolerant needleleaved evergreen and boreal needleleaved summergreen
tree), three types of temperate trees (temperate broadleaved summergreen, temperate shade-intolerant broadleaved
summergreen and temperate broadleaved evergreen tree), three types of tropical trees (tropical broadleaved evergreen tree,
tropical shade-intolerant broadleaved evergreen and tropical broadleaved raingreen tree) and two types of grasses (C3 [cold]
and C4 [warm] grass). LPJ-GUESS also includes plant and soil dynamics of nitrogen (N) and carbon (C) and N-C interactions
for both natural vegetation (Smith et al., 2014; Wårlind et al., 2014) and crops (Olin et al., 2015). N controls plant productivity
due to its limited availability and affects carbon storage in terrestrial ecosystems.

Lindeskog et al. (2013) integrated cropland and pastures and their management as well as vegetation recovery and
succession following cropland abandonment into LPJ-GUESS. This means that the model fully accounts for both land
management and anthropogenic land-use and land-cover changes. Cropland is represented by 11 crop functional types (CFTs),
i.e., temperate cereals, rapeseed, maize, pulses, sugar beet, rice, soybean, sunflower, tropical cereals, peanut, and cassava.
Croplands are harvested each year and the two grass PFTs (C3 and C4 grass) are used as cover crops between harvest and
sowing, and the same two PFTs are used to represent pastures. In any modelled grid cell, LPJ-GUESS simulates the dynamic,
climate-determined structure, composition and functioning of separate stands representing natural vegetation, pasture,
cropland, managed forest and wetlands. The fraction of the grid cell covered by each stand type can change in time, following
external data sets of land-use and land-cover changes. As part of EC-Earth3, LPJ-GUESS uses the gross land-use and land-


cover changes from the LUH2 data set according to observational estimates until 2014 and according to different SSP-scenarios
thereafter (Hurtt et al., 2020).

## 2.2 EC-Earth atmospheric model

In its most simple configuration, the EC-Earth3 ESM is run as an atmospheric model, where the states of the oceans and sea-ice as well as the state of vegetation (see Section 2.1.1) are prescribed according to some external data sets. Also, the atmospheric concentrations of the important GHGs are prescribed according to observations or according to scenarios for the
future. For EC-Earth3, monthly means of sea-surface temperatures (SSTs) and sea-ice conditions for the period 1870-2018 (June) are prescribed according to the PCMDI merged SST data set based on UK MetOffice HadISST (1870-1981 [October]) and NCEP OI v2 thereafter. The concentrations of GHGs, i.e., $CO_2$, methane, nitrous oxide, chlorofluorocarbons, and hydrofluorocarbons, are prescribed according to observations for the historic period 1850-2014 (Meinshausen et al., 2017) and according to various SSP-scenarios thereafter (Meinshausen et al., 2020). Consistent with this, the concentrations of $CO_2$
(together with nitrogen deposition) are prescribed to LPJ-GUESS.

### 2.2.1 Prescribing land-surface conditions

In the simulation with the atmospheric component of EC-Earth3 (see Section 2.3) the land-surface conditions are restricted. As for vegetation (see Section 2.1.1), this means that monthly mean values of the fractions and types of high and low vegetation and the leaf area index for high and low vegetation, respectively, are prescribed from an offline simulation with the land-
surface component HTESSEL+LPJ-GUESS forced with meteorological data from ERA5 (see Section 2.3).

Soil moisture is restricted by nudging the volumetric soil water in the different soil layers towards the 6-hourly values from the offline simulation forced with ERA5. A low-pass filter, i.e., a 121-values running means, had been applied to the 6-hourly values of soil moisture to suppress the variability in the range of sub-daily to intraseasonal time scales. The nudging is done at every time step of the model, and the two nearest 6-hourly values are linearly interpolated to the model time. The
nudging procedure follows the approach outlined in Jeuken et al. (1996), where the model is forced towards the prescribed value via a Newtonian relaxation term, that is the simulated value $x^{sim}$ at the time $t+\Delta t$, with $\Delta t$ being the time step of the simulation, is forced towards the prescribed value for that point in time $x^{pre}(t+\Delta t)$ resulting in a nudged value $x^{ndg}(t+\Delta t)$ according to

$$\frac{X^{ndg}(t+\Delta t) - X^{sim}(t+\Delta t)}{2\Delta t} = \frac{X^{pre}(t+\Delta t) - X^{ndg}(t+\Delta t)}{\tau(X)}$$

with the relaxation time $\tau$ that may depend on the variable x. The nudged value $x^{ndg}(t+\Delta t)$ is then

$$X^{ndg}(t+\Delta t) = \frac{1}{1+\frac{2\Delta t}{\tau(X)}}\left(X^{sim}(t+\Delta t) + \frac{2\Delta t}{\tau(x)}X^{pre}(t+\Delta t)\right)$$


Here, the nudging is only applied for the three lowest soil layers between 7 and 255 cm, with shorter relaxation times for the deeper layers, i.e., 72 hrs for the layer 7-28 cm, 48 hrs for 28-100 cm and 24 hrs for 100-255 cm. The purpose of this design is twofold. Not restricting the uppermost soil layer avoids the introduction of artificial energy fluxes at the land surface under certain meteorological conditions. If, for instance, the model has simulated a rainfall event, which increases the soil moisture,
but the prescribed soil moisture is relatively low, the model would have to evaporate a considerable amount of moisture to reach the low prescribed value. That would introduce strong fluxes of latent heat at the expense of the fluxes of sensible heat that, in turn, affect the conditions in the boundary layer. Choosing varying relaxation times for the different layers means that the more a layer is affected by varying weather conditions the less the soil moisture in that layer is restricted, which reduces



to some extent the moisture fluxes between the layers to reach the prescribed values of soil moisture. In addition, soil moisture
is only nudged, if the ground is not frozen. According to Hauser et al. (2017), this can have a strong effect on the heat fluxes
in the Community Earth System Model and, thus, on the ground heat flux and land-surface temperatures. In the case of, for
instance, higher prescribed soil moisture under freezing conditions the model would have to freeze the additional soil moisture
to reach the prescribed value, which in this case is frozen.

## 2.3 Simulations

This study is based on two simulations. One is an offline simulation with HTESSEL+LPJ-GUESS, the land-surface component
of EC-Earth3, forced by observed meteorological conditions and one is a simulation with the atmospheric component of EC-
Earth3, IFS, where the land-surface conditions are restricted by the offline simulation. Both simulations were performed at the
standard resolution of EC-Earth3, i.e., at T255 spectral resolution and 91 vertical levels or the atmosphere (temperature,
geopotential, divergence, and vorticity) and the N128 linearly reduced Gaussian grid with 256 latitudes and up to 512
longitudes, with the maximum number of longitudes in the tropical belt between about 18° S and 18° N, for other variables.
In both simulations, the anthropogenic land-use and land-cover changes (see Section 2.1) and the concentrations of the
important GHGs (see Section 2.2) were prescribed based on observations until 2014 and according to according to the middle-
of-the road scenario SSP2-4.5 (Fricko et al., 2017) thereafter.

The offline simulation with the land-surface component was forced with the meteorological data from ERA5 (Hersbach
et al., 2020). The meteorological forcing data were prescribed at hourly resolution for the downward longwave and shortwave
radiation at the land surface, rainfall and snowfall, surface pressure as well as temperature, specific humidity, and the horizontal
wind speed near the surface (i.e., the lowest atmospheric model level of ERA5). The simulation, covering the period 1959-
2018, was started in 1959 from a similar offline simulation with HTESSEL+LPJ-GUESS forced with the meteorological
conditions from GSWP3. For the first 20 years of simulation, which were used for adjusting to the new meteorological
conditions, the hourly annual cycles of the meteorological conditions were taken as averages over the four years 1979, 1981,
1982 and 1983 for years without a leap day and over the four years 1980, 1984, 1988 and 1992 for the years with a leap day,
respectively.

In the simulation with the atmospheric component of EC-Earth3 the land-surface conditions were restricted by the data
from the offline simulation with HTESSEL+LPJ-GUESS using the procedure outlined in Section 2.2.1. The simulation
230    covering the period 1970-2017 was started from standard initial conditions for 1970, and the first 9 years of the simulation
allowed the atmospheric model to adjust to the new land-surface conditions. Only the common period for the two simulations,
1979-2017, has been used for the analysis. All the model data used in the analyses performed in this study were interpolated
from the native grid of EC-Earth3 to the N128 regular Gaussian grid with 256 latitudes and 512 longitudes. The offline
simulation with HTESSEL+LPJ-GUESS will be referred to as 'LM' in the figures and in the text, the simulation with the
235    atmospheric component of EC-Earth3 as 'GCM'.

## 3 Observational data

To evaluate the quality of the two simulations various observational data sets have been considered, in some cases different
observational data sets for the same hydrometeorological variables (Table 1). The acronyms in Table 1 will be used when
referring to the specific data sets the figures and in the text. The data have been available for different periods, but here only
240    data within the period of investigation, 1979-2017 (or shorter periods), were considered. Prior to the analysis, all observational
data were interpolated to the N128 regular Gaussian grid.



### 3.1 Surface energy fluxes

For the surface energy fluxes, i.e., net radiation as well as sensible and latent heat fluxes, two versions of the FLUXCOM data (Tramontana et al., 2016; Jung et al., 2019) were considered. These data were downloaded as monthly means on a 0.5 degree regular latitude-longitude grid. To obtain the FLUXCOM data, energy flux measurements from FLUXNET eddy covariance towers are merged with remote sensing and meteorological data by means of machine learning (Tramontana et al., 2016). As for the remote sensing part, data from the MODIS-based remotely sensed land-surface variables are used, while for the meteorological data four different data sets are considered, including the GSWP3 data (Jung et al., 2019). The estimates of the surface energy fluxes have been obtained using different machine learning methods, 9 in the case of the remote sensing data and three when the remote sensing data are combined with meteorological data. Also, three different methods to correct the energy balance are applied for the fluxes of sensible and latent heat. Here, the ensemble means combing all the different methods are used, ranging from three in the case of net radiation in FLUXCOM MET to 27 for the fluxes of sensible and latent heat in FLUXCOM SAT (Table 1). The FLUXCOM data are available for all land areas except for non-vegetated regions, mainly covering Greenland and Antarctica, the Sahara and Taklamakan desert as well as the Arabian Peninsula.

### 3.2 Land-surface temperature

For the land-surface temperature, data from ERA5 were considered (Hersbach et al., 2020). These data were downloaded as 6-hourly data on the native N320 reduced Gaussian grid, and monthly mean values were computed before the interpolation to the N128 regular Gaussian grid. ERA5 is the latest set of global reanalyses prepared by ECMWF, based on a recent version of the IFS, i.e., cycle 41r2, which was operational in 2016. Compared to previous reanalyses, ERA5 has benefited from a decade of developments in model physics, core dynamics and data assimilation as well as from the enhanced horizontal resolution of about 31 km. The ERA5 data are available for all land areas. Here, the land-surface temperature is represented by the so-called 'skin temperature'. The skin temperature is the theoretical temperature of the Earth's surface that is required to satisfy the surface energy balance (Muñoz-Sabater et al., 2021). It represents the temperature of the uppermost surface layer, which has no heat capacity and thus can respond instantaneously to changes in surface fluxes. In HTESSEL, which is used both for ERA5 and in EC-Earth3, the surface energy balance is satisfied independently for each tile (see Section 2.1.1) by calculating its skin temperature (ECMWF, 2016). The skin layer represents the vegetation layer, the top layer of the bare soil or the top layer of the snowpack, respectively.

### 3.3 Surface soil moisture

For surface soil moisture, several data sets were used, i.e., data from the ESA CCI SM project (Dorigo et al., 2017; Gruber at al., 2019) and two versions of the data from the GLEAM project (Miralles et al., 2011; Martens et al., 2017). For both data sets this means the uppermost 10 cm of the soil.

ESA CCI is a satellite-based product of soil moisture, harmonising and merging multiple satellites into three products, one which includes active microwave data only, one with passive microwave data only and a product combining the active and passive microwave products (Gruber et al., 2019). Here, the combined product of version 5.02 is considered, downloaded as daily data on the native 0.25 degree regular latitude-longitude grid. The daily data were aggregated into monthly mean values, accounting for gaps, both in time and space, in the data coverage. In fact, over time the data coverage has increased from 10-20 % of the days per month with valid observations in the 1980s to 50-90% after 2007, except for the high northern latitudes during winter (Dorigo et al., 2016). Valid ESA CCI data are available for all land areas except ice-covered Greenland and Antarctica and areas with tropical rainforests, i.e., Amazonia, Central and coastal Western Africa as well as much of the Maritime continent. The latter is due to the large uncertainty of the estimates in these regions with dense vegetation.

In contrast, the GLEAM (Global Land Evaporation Amsterdam Model) project employs a modelling approach, where a set of algorithms is used to estimate terrestrial evaporation and root-zone and surface soil moisture from satellite data (Miralles





et al., 2011; Martens et al., 2017). In the most recent version 3, GLEAM is used to produce three data sets with different input data, with two of them being used here. GLEAM 3.5a is based on satellite estimates of soil moisture, vegetation optical depth

and snow water equivalent, reanalyses for air temperature and radiation, and a multi-source product for precipitation (Martens et al., 2017). In contrast, GLEAM v3.5b is also based on satellite estimates for air temperature and radiation and a different multi-source product for precipitation. GLEAM data are available for all land areas, but the values are constant in ice-covered Greenland and Antarctica. The data were downloaded as daily data on a 0.25 degree regular latitude-longitude grid and interpolated to the N128 regular Gaussian grid before computing monthly mean values.

**3.4 Precipitation**

The GPCC (Global Precipitation Climatology Centre) has produced several precipitation products, including the Full Data Monthly Product Version 2020, which builds on rain gauges from the GTS-network and on historical data (Schneider et al., 2020). These are about 85,000 stations worldwide with records of 10 years or longer. For individual months, the data coverage varies between 15,000 and more than 50,000 stations. Thus, this data set is the most accurate in situ precipitation reanalysis

data set of GPCC. The data are available for all land areas except Antarctica. The data were downloaded as monthly means on a 0.25 regular latitude-longitude grid.

**4 Quality of the land-surface component**

In this section, the quality of the offline simulation with the land-surface component of EC-Earth3, HTESSEL+LPJ-GUESS (see Section 2.3), is evaluated. If the meteorological forcing from ERA5 represents the 'true' climate, this simulation serves

as a benchmark of the land-surface component. Here, the focus is on the boreal summer season, including June, July, and August.

**4.1 Land-surface temperature**

The simulation with HTESSEL+LPJ-GUESS is characterized by both warm and cold biases compared to ERA5 (Fig. 1b). LM shows warm biases in the northern part of South America, in particular northeast Brazil, in Africa to the south of the equator,

in North America, Northern and Central Europe as well as Central and northeast Asia. Cold biases, on the other hand, are found in the southern part of South America, in Africa to the north of the equator, in western Australia and northwest Asia. Overall, LM is characterized by a small positive bias (Fig. 2a and Table 2).

It should be noted that the land-surface temperatures from ERA5 are reanalyses rather than direct or indirect observational estimates. Also, the fact that the modelling system used to produce ERA5 shares the land-surface model HTESSEL (an updated

version, though) with EC-Earth might cause an underestimation of the biases in LM presented here. The role of vegetation, however, is quite different in ERA5, where the vegetation type and cover as well as land use are time-invariant and surface albedo as well as leaf area index are prescribed by monthly climatology (Muñoz-Sabater et al., 2021). In LM, however, the state of vegetation as well as land use and land cover vary through time (see Section 2.1.2). A serious shortcoming of existing satellite-based data sets for land-surface temperature is that they depend on clear-sky conditions and, thus, do not incorporate

periods with partly or fully cloudy conditions (Johannsen et al., 2019).

**4.2 Surface soil moisture and precipitation**

Compared to ESA CCI, LM is characterized by dry biases in all areas where surface soil moisture is relatively low in the simulation and wet biases in all regions where soil moisture is rather high (Fig. 3a). Thus, dry biases occur in western North America and northeast Brazil, in the northern, southern and much of the eastern parts of Africa, in the Middle East and Central



Asia as well as in Australia (Fig. 3b). Wet biases are found in northern South America, in West and Central Africa, in Central
      Europe and northern Asia as well as in Southeast Asia. The overall negative bias is rather small (Fig. 4a and Table 2).

      The comparison with GLEAM 3.5b, however, gives a quite different picture, with a dry bias of LM over most of the global
      land areas (Fig. S1a). Wet biases are only found in parts of the central tropics and in Southeast Asia. Hence, LM shows an
      overall dry bias in this case (Fig. S1c). This points to some differences between GLEAM 3.5b and ESA CCI, which can be
seen in the differences between these two data sets. GLEAM 3.5b is generally wetter than ESA CCI in regions with relatively
      high surface soil moisture in GLEAM 3.5b and dryer in regions with rather low soil moisture, particularly in Northern Africa,
      in the Middle East and Central Asia as well as in Australia (Fig. S2a). The positive overall difference between the two data
      sets indicates that soil moisture is generally higher in GLEAM 3.5b than in ESA CCI (Fig. S2b). Apparently, LM and GLEAM
      3.5b reveal similar tendencies when compared to ESA CCI, except for a general tendency of higher soil moisture in GLEAM
3.5b, which explains the almost uniform dry bias in LM against GLEAM 3.5b. Using different meteorological forcing data for
      GLEAM (see Section 3.3) has only a small effect on surface soil moisture, as indicated by the difference between GLEAM
      3.5a and GLEAM 3.5b. Using reanalyses of air temperature and radiation and a different multi-source product for precipitation
      in GLEAM 3.5a results in slightly higher surface soil moisture in most areas except in the western United States and Central
      Africa (Fig. S3a), resulting in a positive overall difference (Fig. S3b).

Here, ESA CCI has been chosen as the primary reference data set because it is a pure remote-sensing product, while
      GLEAM uses a modelling approach so that the estimates might be affected by the characteristics of the model. There have
      been efforts to evaluate the quality of the soil moisture products (typically focusing on the temporal dynamics) during the
      development for each of the two data sets. Beck et al. (2021) evaluated 18 different soil moisture products against in situ
      measurement in Europe and, especially, the USA for 2015-2019. Focussing on the temporal dynamics, the authors found
differences in the temporal correlations between the different data sets, with GLEAM doing better than ESA CCI. The
      performance of ESA CCI was generally worse for the high-frequency variations and for the sites with arid or cold conditions.
      Nevertheless, there were also sites, where ESA CCI showed a better performance than a merged multi-model product at some
      sites despite a worse performance at other sites not far away. The results presented in Beck et al. (2021) do, however, not give
      any specific information on the quality of the long-term means of surface soil moisture of the data sets considered here.

One of the important drivers of soil moisture is precipitation. Therefore, the quality of the precipitation in the
      meteorological forcing from ERA5 may affect the quality of the soil moisture simulated by HTESSEL+LPJ-GUESS. Relative
      to GPCC, precipitation in ERA5 is too low in the central tropics and in eastern USA and too high in the mid- and high latitudes
      as well as in Southeast Asia (Fig. 5b). Overall, precipitation is slightly stronger in ERA5 than for GPCC (Fig. 6a and Table 2).
      Nogueira (2020) found a similar pattern of the bias in annual mean precipitation of ERA5 against the GPCP data. A comparison
with the bias in surface soil moisture (see Fig. 3b) shows that it is mainly in Southeast and in northern Asia that the excess in
      precipitation might have contributed to the wet bias in soil moisture. To reduce the effects of potential biases in precipitation,
      Muñoz-Sabater et al. (2021) scaled the precipitation from ERA5 with the monthly-mean climatology of GPCP when they
      forced an updated version of HTESSEL, CHTESSEL, to produce the ERA5-Land reanalyses. In their evaluation against in
      situ measurements they did not find any noticeable differences between the surface soil moisture in ERA5 and ERA5-Land
for selected sites in Europe and Africa, but a reduced bias in Australia, possibly because all the sites are located on the coast,
      where the much finer resolution of ERA5-Land may have an effect.

### 4.3 Surface energy fluxes

      The energy balance at the land surface is determined by, on one hand, the incoming and outgoing fluxes of shortwave (SW)
      and longwave radiation (LW) and the fluxes of sensible (SHF) and latent heat (LHF) as well as the ground heat flux (G) on
the other (e.g., Teuling et al., 2010), that is



$$SW_\downarrow + SW_\uparrow + LW_\downarrow + LW_\uparrow = SHF + LHF + G$$

The incoming and outgoing shortwave radiation are connected via the surface albedo α as


$$SW_\uparrow = -\alpha SW_\downarrow$$

and the longwave outgoing radiation relates to the land-surface temperature (T) via


$$LW_\downarrow = \varepsilon \sigma T^4$$

with the thermal emissivity ε and the Stefan-Boltzmann constant σ. Omitting the ground heat flux and combining the four components of the radiative fluxes into the net radiation ($R_n$) reaching the ground this results in


$$-R_n = SHF + LHF$$

using the convention that upward fluxes are positive and downward fluxes negative, respectively. That is, the incoming net radiation at the land surface is separated into the sensible and latent heat fluxes into the atmosphere.

Compared to FLUXCOM SAT, LM is characterized by an underestimation of the incoming net radiation over large parts
of the global land areas, in particular South America and Africa south of the equator but also in the mid- and high latitudes of the Northern Hemisphere (Fig. 7b). LM overestimates net radiation mainly in the western USA, Central Asia and western Australia. This distribution of the biases results in an overall underestimation of net radiation in the simulation (Fig. 8a and Table 2). LM is also characterized by a marked underestimation of the sensible heat flux in the mid- and high latitudes of the Northern Hemisphere, in Amazonia and southern South America as well as in Southeast and East Asia (Fig. 9b). The fluxes
of sensible heat are overestimated in the western USA, northeast Brazil and Central Asia, in the southern Sahel region, East Africa and the northern part of Southern Africa as well as in northern Australia. Thus, the distribution of the biases illustrates the model's tendency to underestimate the fluxes of sensible heat in regions where they are relatively weak and overestimate them in regions where they are relatively strong (Fig. 9a). Considering all land areas, sensible heat fluxes are noticeably underestimated (Fig. 10a and Table 2). For latent heat, LM also shows a slight overall underestimation of the fluxes (Fig. 12a
and Table 2), which is the result of rather strong negative biases in northeast Brazil, in Southern and East Africa as well as in northern Australia (Fig. 11b). These are also the regions with the most distinct positive biases in sensible heat fluxes (see Fig. 9b). The regions with a pronounced overestimation of the latent heat fluxes, on the other hand, correspond by and large to the areas with too weak sensible heat fluxes, except for the very northern part of Asia, where the fluxes of latent heat are slightly overestimated but the sensible heat fluxes strongly underestimated. In contrast to the sensible heat flux, LM doesn't show a
general tendency of overestimating the fluxes of latent heat in regions where the fluxes are rather strong and of underestimating them where they are relatively weak. The situation with too strong fluxes of latent heat is primarily found in northeast Brazil as well as in Southern and East Africa and the situation with too weak fluxes in Amazonia, eastern North America and Western Europe as well as in the southern part of West Africa, in Central Africa as well as in Southeast and East Asia (Fig. 11a). The geographical distribution of the biases in the evaporative fraction (Fig. 13b) is very similar to the fluxes of latent heat (see Fig.
11b), indicating that the model biases in the evaporative fraction are mainly related to the biases in latent heat flux. In the high northern latitudes of Europe and Asia, however, the marked underestimation of the sensible heat flux (see Fig. 9b) strongly contributes to the overestimation of the evaporative fraction (defined as the ration between the latent heat flux [LHF] and the total heat flux defined as the sum of the latent and sensible heat flux [LHF+SHF]) in these areas. Overall, LM is characterized by a slight overestimation of the evaporative fraction (Fig. 14a), even though the negative biases in especially northeast Brazil



as well as in Southern and East Africa are rather strong. The geographical distribution of the biases in the fluxes of latent heat have similarities with the bias of surface soil moisture with respect to ESA CCI (see Fig. 3b) but not compared to GLEAM 3.5b (see Fig. S1a), suggesting that ESA CCI in fact is  be the better choice as a reference data set for surface soil moisture.

Using a different version of FLUXCOM, in this case based on GSWP3 as meteorological data, as reference data reveals some differences compared to FLUXCOM SAT. For net radiation, the negative biases are smaller in northeast Brazil and

Southern Africa but larger in the high northern latitudes (Fig. S4a). In some regions the biases have opposite signs, such as in Central Africa, where LM is characterized by an overestimation of net radiation against FLUXCOM MET but an underestimation compared to FLUXCOM SAT (see Fig. 7b). The slight overall underestimation of net radiation is about the same for both reference data sets (Fig. S4c). For the flux of sensible heat, using FLUXCOM MET reveals a negative bias in Central and Southern Africa (Fig. S5a) in contrast to the positive bias against FLUXCOM SAT. Also, the overestimation of

the sensible heat flux in northeast Brazil is considerably weaker with respect to FLUXCOM MET and the underestimation in the other parts of South America is stronger. The latter is also the case in Australia. These discrepancies lead to a somewhat stronger overall negative bias in sensible heat flux for FLUXCOM MET (Fig. S5c). At the same time the overall negative bias in latent heat flux for FLUXCOM SAT is turned into a slight positive bias (Fig. S6c). This is mainly because the underestimation in the fluxes of latent heat in northeast Brazil is considerably weaker and the negative bias in Southern Africa

has almost disappeared (Fig. S6a). On the other hand, the overestimation of latent heat fluxes in the high northern latitudes is reduced, notably in Europe and Asia. The discrepancies between the biases in the surface energy fluxes with respect to the two reference data sets also affect the bias in the evaporative fraction. That is, the underestimation of the evaporative fraction in northeast Brazil is markedly reduced and turned into a slight overestimation in Southern Africa (Fig. S7a). At the same time, there is a notable overestimation in Central Asia and to some extent in the western USA.

The marked effects of the specific version of the FLUXCOM data on the biases suggest some systematic differences between the two data sets. FLUXCOM MET is, for instance, characterized by stronger net radiation in the high northern latitudes, in western USA as well as in the Middle East and much of Australia (Fig. S8a). Net radiation is weaker in FLUCXOM MET in eastern USA and most of South America, in southern Europe and in most of Africa south of the Sahel, in Southeast and East Asia as well as on the Maritime continent. The fluxes of sensible heat are rather strong in the high northern latitudes

in Europe and Asia as well as Alaska, in most of South America, in Africa south of the Sahel as well as on the Maritime continent and in Australia (Fig. S9a). Sensible heat fluxes are weaker in North America except Alaska as well as Europe and Asia in a latitudinal belt between about 10 and 50 °N. For the fluxes of latent heat, the differences between the two versions of the FLUXCOM data set have the opposite sign than the differences in sensible heat flux except for Alaska, the high northern latitudes in Europe and Asia and, to some extent, southern Australia, where both fluxes are stronger in FLUXCOM MET (Fig.

S10a). The geographical distribution of the differences in the evaporative fraction (Fig. S11a) are comparable to the latent heat fluxes, indicating that the differences in latent heat flux give the main contribution to the discrepancies in the evaporative fraction.

### 4.4 Regional perspective

The updated IPCC reference regions (Iturbide et al., 2020) are used for adding a regional perspective to the analyses undertaken

in this study. Here, 44 land regions, excluding the two land regions in Antarctica are considered. See Table S1 for a list of acronyms and the names of these regions and Figure 1b in Iturbide et al. (2020) for a map displaying the regions. Area mean values were computed for these regions, accounting for the actual land area represented by a grid box. To facilitate the work with various data sets covering different time periods (Table 1), the area mean values were computed for the common period of all observational data sets considered here, 2003-2014.

In Figure 15, FLUXCOM SAT, ERA5 and ESA CCI were used when computing the differences in the regional means between the two simulations and the reference data sets. According to this, LM has a warm bias in land-surface temperatures



in most of the regions, with the exceptions of southwestern, south-eastern and southern South America, the Sahara and north-eastern Africa, several regions in subtropical Asia and most regions in Australia (Fig. 15a). LM underestimates surface soil moisture in most of the regions, but overestimates it in quite a few regions, typically located in the high northern latitudes and in Asia. The biases in soil moisture are clearly associated with corresponding biases in the fluxes of latent heat, that is generally with too strong and too weak latent heat fluxes in regions with too high and too low soil moisture, respectively. The biases in latent heat fluxes, in turn, are linked to biases in the fluxes of sensible heat with the opposite sign, such that in most regions positive biases in latent heat flux are linked to negative biases in sensible heat flux and vice versa. In most of the regions the biases in latent heat flux are associated with corresponding biases in the evaporative fraction. The regional differences also clearly indicate that generally LM underestimates the incoming net radiation. Only in western and central North America, western Siberian and central Australia is the net radiation overestimated by the model.

Using different reference data sets for surface soil moisture (GLEAM 3.5b instead of ESA CCI) has a profound effect on the differences in the regional means between and LM and the observational data set. That is, LM underestimates soil moisture in almost all regions except for northern Europe, the Tibetan Plateau and Southeast Asia (Fig. S12a). This affects the connection with the bias in latent heat flux, where only about 25 regions are characterized by corresponding biases with too weak and too strong fluxes of latent heat in regions with too low and too high soil moisture, respectively. For the two other components of the surface energy fluxes, the overall picture is the same for FLUXCOM MET as for FLUXCOM SAT with an underestimation of the incoming net radiation and, to a lesser extent, of the sensible heat flux. Again, the biases in sensible heat flux typically correspond to biases in the fluxes of latent heat of the opposite sign, and the biases in latent heat flux are linked with corresponding biases in the evaporative fraction. Despite the similar overall tendencies, some discrepancies with respect to the two versions of the FLUXCOM data set can be found. Using FLUXCOM MET results, for instance, in more regions where the incoming net radiation is overestimated, i.e., eastern North America, the Caribbean and north-western South America, Western and Central Europe, Central Africa, the Tibetan Plateau as well as East and South Asia. Another marked difference is the underestimation of the sensible heat flux in the southern part of Africa with respect to FLUXCOM MET compared to an overestimation when FLUXCOM SAT is used.

## 5 Role of the atmospheric coupling

In this section, the quality of the simulation with the atmospheric component of EC-Earth3 with the land-surface conditions prescribed from the offline simulation (see Section 2.3) is assessed. As the land-surface conditions are prescribed, this analysis illustrates the role of the atmospheric coupling for the biases in EC-Earth3. Feedbacks from the atmosphere have a very limited effect on the soil moisture in the three lowest layers (nudged) and no effect on vegetation (prescribed) but on the surface energy balance through the energy fluxes at the land surface. The impact of the atmosphere is depicted by the differences between the simulation with EC-Earth3 and the simulation with HTESSEL+LPJ-GUESS. This simulation with the atmospheric component of EC-Earth3 serves as a benchmark of the EC-Earth3 ESM (not incorporating the biogeochemical couplings, though), except for the potential compensation of errors when the land surface is fully coupled to the atmosphere.

### 5.1 Land-surface temperature

The simulation with EC-Earth3 is characterized by both warm and cold biases compared to ERA5 (Fig. 1c). Consistent with the simulation with HTESSEL+LJP-GUESS, GCM shows warm biases in northeast Brazil, the United States and Central Asia. Cold biases, on the other hand, are not only more pronounced in the simulation with EC-Earth3, but also cover a larger fraction of the globe, e.g., the northern part of North America, northern and Central Europe, northeast Asia, Central Africa as well as India and Southeast Asia. This means that the coupling with the atmosphere enhances the warm biases of LM in northeast Brazil, the south-western part of North America and Central Asia but converts them into cold biases in the northern part of



North America, northern and Central Europe, northeast Asia, Central Africa as well as India and Southeast Asia (Fig. 1d). Thus, the small positive overall bias in LM is changed into a marked negative overall bias by the coupling with the atmospheric component of EC-Earth3, and the overall negative difference between the two simulations exceeds the overall bias of GCM

(Figs. 2b, c and Table 2). The distributions of the differences with ERA5 clearly show that the local biases in land-surface temperature become more pronounced in GCM with a range of -6 to 6 °C in GCM (Fig. 2b) than in LM with a range of -4 to 4 °C (Fig. 2a).

### 5.2 Surface soil moisture and precipitation

The coupling with the atmospheric component of EC-Earth3 has only a small effect on the bias of surface soil moisture. The

distribution of the wet and dry biases for GCM (Fig. 3c) is very similar to the distribution of the biases for LM (Fig. 3b) and the local differences between GCM and LM are rather small (Fig. 3d). This indicates that surface soil moisture is predominantly controlled by the moisture conditions in the lower soil layers (that have been constrained in GCM) and to a lesser extent by the coupling with the atmosphere. The distribution of the differences between the two simulations reveals that the coupling with the atmosphere enhances some of the biases that already occur in the simulation with HTESSEL+LPJ-GUESS, i.e., the

wet biases in the high northern latitudes and India and the dry biases in southern part of North America and Central Asia (Fig. 3d). The coupling with the atmosphere changes the small negative overall bias in LM to a small positive overall bias in GCM (Figs. 4a, b and Table 2).

Most of the differences in surface soil moisture between the two simulations can be explained by the differences in precipitation with wet and dry biases in corresponding regions (Figs. 3d, 5d). Notable exceptions are the negative differences

in precipitation in northeast Southeast Asia (Fig. 5d) with positive differences in surface soil moisture in these regions (Fig. 3d). GCM underestimates precipitation in some parts of the tropics and in the subtropics and overestimates it in the extra-tropics and other parts of the tropics (Fig. 5c). Overall, GCM is characterized by too much precipitation (Fig. 6b) but not as much as LM (i.e., ERA5), resulting in a negative difference between GCM and LM (Table 2). This confirms the observation that surface soil moisture is for the most part controlled by the moisture conditions in the lower soil layers.

### 5.3 Surface energy fluxes

Like the simulation with HTESSEL+LPJ-GUESS, GCM underestimates the incoming net radiation over large parts of the global land areas, e.g., in northeast Brazil and Africa south of the equator as well as in the Northern Hemisphere extra-tropics (Fig. 7c). GCM overestimates net radiation in the southern part of North America and the northern part of South America, in southern Europe and Central Asia as well as at the Horn of Africa. Compared to LM, this means a reduction of the overall

negative bias when interacting with the atmosphere (Figs. 8a, b and Table 2). This is partly due to a weakening of the negative biases or a strengthening of the positive biases, but also to a shift from negative to positive biases is some areas, e.g., the south-eastern United States, the northern part of South America, Southern Europe and at the Horn of Africa. In contrast to the net radiation, the geographical distribution of the biases in the fluxes of sensible heat is very similar for the two simulations (Figs. 9b, c). Nevertheless, the fluxes of sensible heat are generally stronger in the simulation with the atmospheric component of

EC-Earth3 (Fig. 9d and Table 2). Notable exceptions with weaker sensible heat fluxes are found for the northern part of North America and northeast Asia as well as the southern part of Central Africa. In contrast to LM, the simulation with the atmospheric component of EC-Earth3 is characterized by an underestimation of the fluxes of latent heat in northeast Asia (Fig. 11c). Other than that, the geographical distributions of the biases are similar for the two simulations (Figs. 11b, c). The coupling with the atmosphere reduces the fluxes of latent heat not only in northeast Asia but also in the northern part of North America,

most parts of Europe and Asia as well as in Central Africa (Fig. 11d). The latent heat fluxes are stronger in South America (especially in the northern part), the south-eastern part of North America as well as near the Horn of Africa, in parts of South Asia and Australia (Fig. 11d). Overall, the fluxes of latent heat are slightly weaker in GCM than in LM (Fig. 12c and Table



2). The coupling with the atmosphere has only a small effect on the geographical distribution of the biases in the evaporative fraction (Figs, 13b, c) but leads to a reduction of the evaporative fraction in many parts of the global land area and to an
increase in a few regions, e.g., the southern part of the Arabian Peninsula as well as parts of South Asia and Australia (Fig. 13d). This results in a slight reduction of the overall value of the evaporative fraction in the simulation with the atmospheric component of EC-Earth3 (Fig. 14c). In summary, the coupling with the atmosphere leads to an increase of the incoming net radiation at the land surface, resulting in a somewhat stronger increase in the sensible heat flux, which is partly compensated by a decrease in the latent heat flux. The overall biases for the three types of surface energy fluxes are considerably reduced
when HTESSEL+LPJ-GUESS is interacting with the atmosphere, but the corresponding root-mean-square (RMS) deviations are slightly lager for the simulation with the atmospheric component of EC-Earth3 (Table 2).

### 5.4 Regional and global perspective

Compared to the simulation with HTESSEL+LPJ-GUESS, the regional means of the biases for the atmospheric model show more regions where the incoming net radiation is overestimated (Fig. 15b) and a strengthening of the positive biases in the
four regions where LM overestimates the net radiation (Fig. 15a). For the fluxes of sensible and latent heat there is not much of a change in the sign of the biases in the individual regions, but a general tendency of a strengthening of the positive biases and a weakening of the negative biases for the fluxes of sensible heat. The coupling with the atmosphere has a notable effect on the regional means of the land-surface temperatures with more regions with a cold bias and stronger cold biases in regions where LM is characterized by a cold bias. The deviations between the two simulations become clearly visible in the regional
means of the differences between GCM and LM. That are, for instance, the stronger incoming net radiation in GCM resulting in stronger fluxes of sensible heat (Fig. 16). Only in the high latitudes of the Northern Hemisphere (i.e., Greenland/Iceland, northwest and northeast North America, the Russian Arctic as well as the Russian Far East), in southern South America, Tibet, the western part of Southern Africa and New Zealand are the sensible heat fluxes stronger in LM, consistent with stronger net radiation in these regions. Although the differences in the fluxes of latent heat follow to some extent the regional differences
in net radiation, there are some contrasts, with weaker latent heat fluxes in regions with stronger net radiation in some regions. For the evaporative fraction, on the other hand, there is a distinct tendency of a reduction due to the coupling with the atmosphere. Only in the Caribbean, southern South America, the Arabian Peninsula, Tibet and New Zealand is the evaporative fraction stronger in GCM. The land-surface temperatures are cooler in the simulation with the atmospheric component of EC-Earth3 for most of the regions with several exceptions, i.e., western and central North America, northern Central America,
north-eastern South America, the Mediterranean region, western and central East Asia, the Arabian Peninsula as well as northern, eastern and southern Australia. Surface soil moisture is reduced in GCM for most of the regions, but the regional mean differences between the two simulations are small (typically between -2 and +2%) compared to the regional mean biases (Fig. 15).

A Taylor diagram is used to illustrate how well the global patterns obtained from the two simulations match the reference
data (Taylor et al., 2001). Figure 17 thus shows the correlations between the simulated and the reference fields (straight lines), the normalized RMS differences for the simulated (ordinate) and the reference fields (abscissa) as well as the normalized RMS deviation between the simulated and the reference fields (half circles) for the two simulations, LM and GCM. According to the diagram, the simulation with HTESSEL+LPJ-GUESS has better skills than the simulation with the atmospheric component of EC-Earth3 for all variables except precipitation. For precipitation all three aspects are better for GCM, while for the surface
exergy fluxes and the land-surface temperatures all three aspects are superior for LM. For surface soil moisture, on the other hand, the correlation with the reference field is slightly higher for GCM, while the RMS difference is lower for LM. The deviations between the two simulations are relatively large for the net radiation, sensible heat flux and land-surface temperatures and small for the latent heat flux, surface soil moisture and precipitation. For both simulations the skills are particularly low for the sensible heat flux and surface soil moisture.



## 6. Discussion

The quality of the simulation with HTESSEL+LPJ-GUESS can be affected by the choice of the data that were used as meteorological forcing, in this case the ERA5 reanalyses. ERA5 is the most recent set of global reanalyses (ensuring consistency between the meteorological variables), widely used (https://doi.org/10.24381/cds.adbb2d47) as well as well-documented and evaluated (Hersbach et al., 2020). Although being a good presentation of the near-surface climate, ERA5 may have some biases such as in precipitation as documented here (see Section 4.2) or in Nogueira et al. (2020) that can affect the simulation of the land-surface conditions by HTESSEL+LPJ-GUESS.

The estimates of the quality of the model configurations, i.e., the land-surface component HTESSEL+LPJ-GUESS and the atmospheric component of EC-Earth3, depends to some extent on the choice of the reference data sets. For some of the variables, i.e., surface soil moisture and the surface energy fluxes, several data sets have been used and the choices of a specific reference data set for the quality assessment have been discussed here. A pronounced difference was found for the surface soil moisture, where using the model-based GLEAM data set resulted in a relatively strong dry bias for both model configurations (see Section 4.2). The general intention was to focus on information based on rain-gauge or on remote sensing products, e.g., for soil moisture and as meteorological forcing for the surface energy fluxes. But as estimates of land-surface temperatures based on satellite data depend on clear-sky conditions (Johannsen et al., 2019), model-based estimates from ERA5 were used. Given the fact that ERA5 and the EC-Earth ESM share the HTESSEL land-surface model, that may lead to somewhat conservative estimates of the biases in land-surface temperature.

Furthermore, only one season, i.e., the boreal summer season comprising June, July and August, is analysed in this study. A particular season rather than the annual time range are considered because combining different seasons could lead to somewhat misleading results as the biases for different climatological seasons might compensate each other. The boreal summer season is chosen because it is the growing season in the Northern Hemisphere and, thus, the season when vegetation is most important from a global perspective given the uneven distribution of the land areas between the two hemispheres. With large parts of the high northern latitudes being covered by snow and the reduced or absent incoming solar radiation at these latitudes the overall picture might be quite different for boreal winter.

In this study, the land-surface component of EC-Earth3 is considered as a whole, including both the land-surface scheme HTESSEL and the dynamic vegetation model LPJ-GUESS. This is because the intention is to assess a) the quality of the land-surface component and b) the effects of the coupling with the atmosphere (where both the land surface and vegetation are important) and not to validate each model component by itself. The latter would require additional simulations, additional data sets and further analysis and, thus, would go beyond the scope of this paper.

## 7. Summary and conclusions

The aim of this study is twofold, first to evaluate the quality of simulation of surface climate by the land-surface component of the EC-Earth3 ESM, HTESSEL+LPJ-GUESS, and second to assess the role of the coupling of the land surface with the atmosphere for the simulation of the surface climate in EC-Earth3 is assessed. As variables characterizing the surface climate land-surface temperature, several components of the surface energy fluxes (net radiation, sensible and latent heat flux, evaporative fraction) and surface soil moisture have been included.

### 7.1 Summary

The land-surface component of EC-Earth3 is characterized by marked regional biases in the aspects of surface climate considered here. These are, for instance, too warm land-surface temperatures in the tropics and in the mid- and high latitudes of the Northern Hemisphere, resulting in an overall warm bias. Surface soil moisture, on the other hand, is characterized by a dry bias in the subtropics and parts of the extra-tropics and a wet bias in the tropics and the eastern part of Asia, resulting in a



slightly negative overall bias. The incoming net radiation is underestimated by the model over much of the global land area, causing a negative overall bias. For the fluxes of sensible heat, the model also shows a negative overall bias with a clear tendency to underestimate the sensible heat fluxes in regions, where they are relatively strong, and underestimate them in regions where they are rather weak. The biases in the fluxes of latent heat generally correspond to the biases in the sensible heat fluxes with an underestimation of the fluxes of latent heat in regions where the sensible heat fluxes are too strong and an

overestimation in the regions where the sensible heat fluxes are too weak. This means that the land-surface component overestimates the fluxes of latent heat in the high latitudes of the Northern Hemisphere where the latent heat fluxes generally are relatively weak.

The coupling with the atmosphere leads to somewhat stronger biases in the aspects of surface climate as indicated by the
estimates of the RMS deviation between the simulations and the reference data sets. Only for the land-surface temperature the RMS is markedly stronger for the simulation with the atmospheric component of EC-Earth3 than for the simulation with the land-surface component. Land-surface temperature is also one of the climate variables where the coupling with the atmosphere changes the sign of the overall bias from a warm overall bias in the simulation with the land-surface component to a considerable cold bias in the atmospheric component of EC-Earth3. The other variable is surface soil moisture for which the
coupling with the atmosphere changes a dry overall bias of the land-surface component into a wet bias. Analysing the correspondence between the global patterns for the simulations and the reference data reveals relatively large effects of the atmospheric coupling on land-surface temperature as well as net radiation and sensible heat flux but small effects on surface soil moisture and latent heat flux.

### 7.2 Conclusions

The results of this study show that even under 'true' climate conditions the land-surface component of the EC-Earth3 ESM is characterized by notable biases in land-surface temperatures, surface soil moisture and surface energy fluxes, also affecting the partition between sensible and latent heat fluxes. These biases in the land-surface conditions would result in biases in near-surface climate and, possibly, by affecting the stability in the atmospheric boundary layer cause biases in cloud cover and precipitation. The interactions with the atmosphere as simulated by EC-Earth3 change these biases, such as introducing a
marked overall cold bias in the land-surface temperatures and an overall wet bias in surface soil moisture. Also, the interactions with the atmosphere affect the partition of the surface energy fluxes leading to a general increase in the evaporative fraction. Thus, the interactions with the atmosphere alter the biases in the land-surface component and possibly the biases in near-surface climate and in the atmospheric boundary layer.

The biases in the land-surface component reveal characteristic geographical distributions of notable negative or positive
deviations from the reference data, and the overall structures of these regional patterns are typically only slightly affected by the interactions with the atmosphere. In a local to regional perspective, therefore, the quality of the land-surface component is important for the biases in land-surface conditions and, thus, the local or regional biases in the simulated climate. This illustrates the need for improving the quality of the land-surface component of EC-Earth3 parallel with other components of the ESM. An improved representation of the land-surface conditions would not only contribute to more realistic climate
simulations but also result in more reliable future climate scenarios (Van den Hurk et al., 2016).

### Code availability

In this study, the cdo software (https://code.mpimet.mpg.de/projects/cdo) has been used for the data handling and standard methods have been applied for the analyses (code mainly in Fortran). The code for the analyses can be made available by the author on request.



**Data availability**

The various observational data sets used in the study can be downloaded from the relevant sites. The data from the two simulations (seasonal mean values) can be made available by the author on request.

**Author contribution**

The author developed and implemented the code for nudging soil moisture in the EC-Earth3 ESM and performed the two 655 simulations with the different configurations of EC-Earth3. The author also performed the analyses and prepared the manuscript.

**Competing interests**

The author declares that he hasn't any competing interests.

**Acknowledgements**

The research presented in this paper is a contribution to the Swedish strategic research area Modelling the Regional and Global Earth System, MERGE. Access to the computer facilities at the European Centre for Medium-Range Weather Forecasts (ECMWF) is granted through the special project on "Land surface - climate interactions in the EC-Earth ESM: their role for climate variability and contribution to future climate change". Thanks to the Max Planck Institute for Biogeochemistry for providing the FLUXCOM data, the ESA CCI SM project for providing the ESA CCI data, the GLEAM project for providing 665 the GLEAM data, ECMWF for providing the ERA5 data and the German Weather Service for providing the GPCC data. Also, thanks to José Gutiérrez for the shapefiles for the IPCC reference regions. Special thanks to Emanuel Dutra and Etienne Tourigny for preparing the offline version of the land-surface component of EC-Earth3, HTESSEL+LPJ-GUEESS, and therewith enabling this study.

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

**Tables**


| Variables | Data set | Period | Remarks | Key references |
|---|---|---|---|---|
| Net radiation, sensible and latent heat fluxes | FLUXCOM SAT | 2001-2015 | Remote sensing data only | Tramontana et al. (2016), Jung et al. (2019) |
| | FLUXCOM MET | 1979-2014 | Meteorological forcing from GSWP3 in addition to remote sensing data | |
| Land-surface temperature | ERA5 | 1979-2017 | Extracted from the MARS archive | Hersbach et al. (2020) |
| Surface soil moisture | ESA CCI | 1979-2017 | Version 05.2 | Dorigo et al. (2017), Gruber et al. (2019) |
| | GLEAM 3.5b | 2003-2017 | Remote sensing data only | Miralles et al. (2011), Martens et al. (2017) |
| | GLEAM 3.5a | 1980-2017 | Reanalysis data in addition to remote sensing data | |
| Precipitation | GPCC | 1979-2017 | Full data monthly version 2020 | Schneider et al. (2020) |

**Table 1:** Overview of the observational data sets for different hydrometeorological variables used as reference data.

| Differences | LST [°C] | | SM [%] | | PRE [mm/day] | | RAD [W/m$^2$] | | SHF [W/m$^2$] | | LHF [W/m$^2$] | |
|---|---|---|---|---|---|---|---|---|---|---|---|---|
| LM-OBS | 0.10 | 0.97 | -0.11 | 8.54 | 0.29 | 1.44 | -7.18 | 14.74 | -10.53 | 21.92 | -1.50 | 19.43 |
| GCM-OBS | -0.42 | 1.78 | 0.14 | 8.66 | 0.18 | 1.37 | -3.21 | 16.65 | -6.23 | 22.33 | -3.01 | 19.96 |
| GCM-LM | -0.52 | 1.55 | 0.18 | 1.86 | -0.12 | 1.43 | 3.94 | 12.20 | 4.20 | 9.36 | -1.36 | 8.41 |

**Table 2:** Mean differences and the root-mean-square deviation between LM and the primary observational data sets (LM-OBS), between GCM and the observational data (GCM-OBS) as well as between the two simulations (GCM-LM) for land-surface temperature (LST), surface soil moisture (SM), precipitation (PRE), net radiation (RAD) as well as sensible (SHF) and latent heat flux (LHF). Periods considered are 1979-2017 for LST, SM and PRE and 2001-2015 for RAD, SHF and LHF, respectively. Note that the difference between the two simulations does not necessarily harmonize with the corresponding differences between GCM-OBS and LM due to additional grid points
for GCM-LM for all variables except LST and PRE.

**Figures**





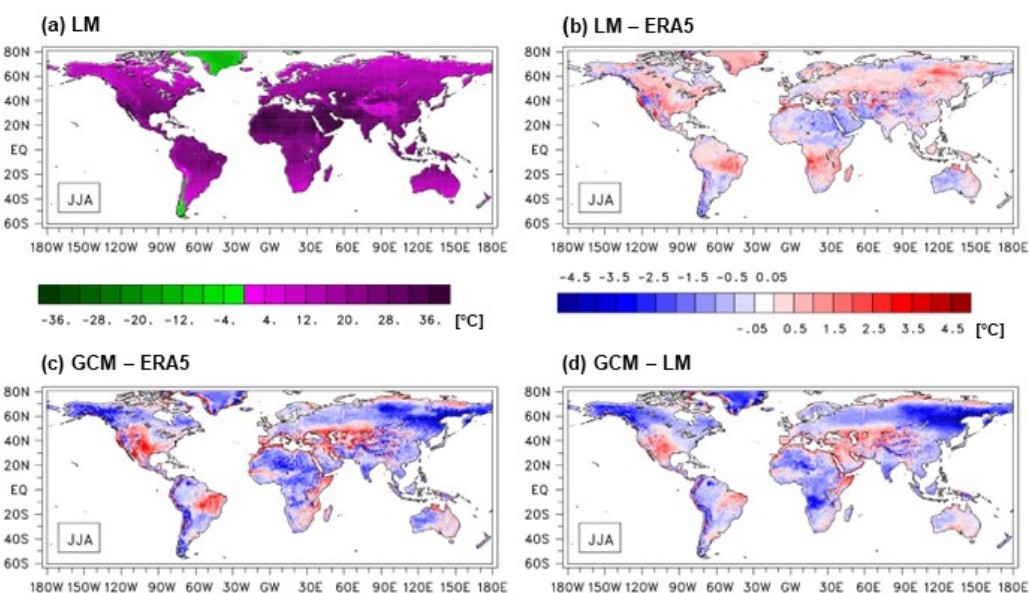

**Figure 1:** Seasonal (June through August; JJA) mean values of land-surface temperatures for 1979-2017 for the simulation with the land
component forced with ERA5 (LM) (a) as well as the differences (b) between LM and ERA5 (LM – ERA5), (c) between the simulation with
the atmospheric component of EC-Earth with the land-surface conditions prescribed from LM (GCM) and ERA5 (GCM – ERA5) and (d)
between GCM and LM (GCM – LM). Units are °C; the contour interval is 2 °C in (a) and 0.5 °C in (b-d), respectively.










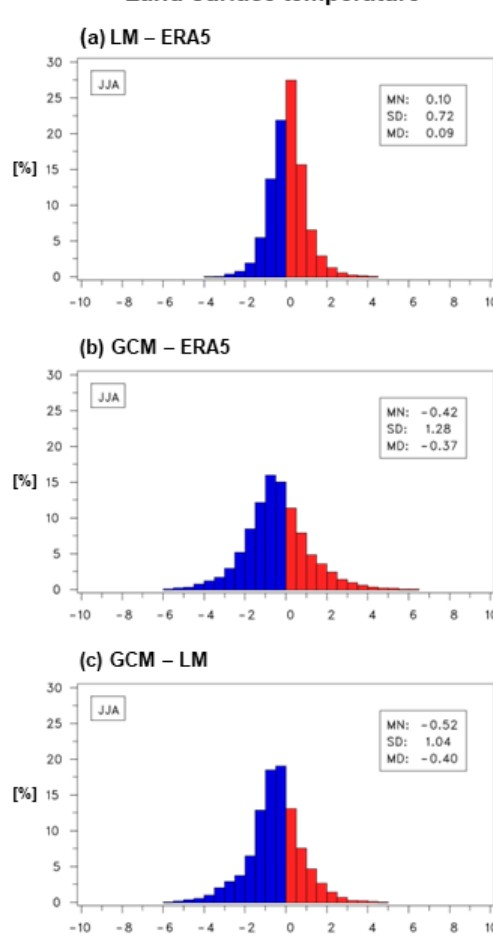


**Figure 2:** Distributions of the area-weighted seasonal mean differences of land-surface temperatures (a) between LM and ERA5 (LM – ERA5), (b) between GCM and ERA5 (GCM – ERA5) and (c) between GCM and LM (GCM – LM). Units are °C on the abscissa and % on the ordinate, respectively. The mean, standard deviation and median of the distribution are given in the box; units are °C.






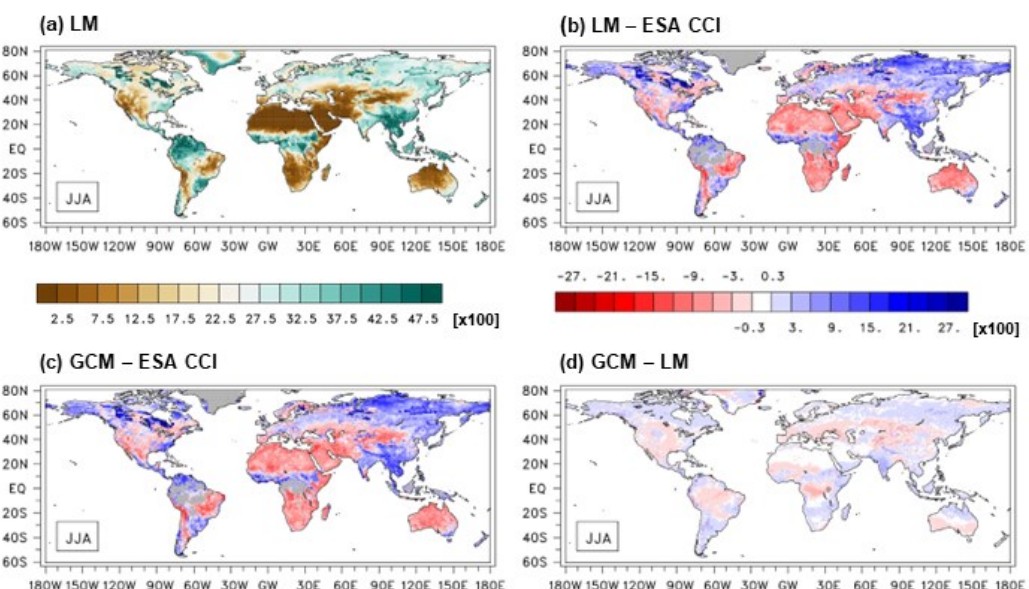

**Figure 3:** Seasonal mean values of surface soil moisture for 1979-2017 for LM (a) as well as the differences (b) between LM and ESA CCI (LM – ESA CCI), (c) between GCM and ESA CCI (GCM – ESA CCI) and (d) between GCM and LM. Units are $m^3/m^3$ (x100); the contour interval is 2.5 $m^3/m^3$ (x100) in (a) and 3 $m^3/m^3$ (x100) in (b-d), respectively. Note the swap of the colour scale with red colours indicating small values (a) or negative differences (b-d) and blue colours indicating high values (a) or positive differences (b-d). Missing data in ESA CCI are marked in grey.





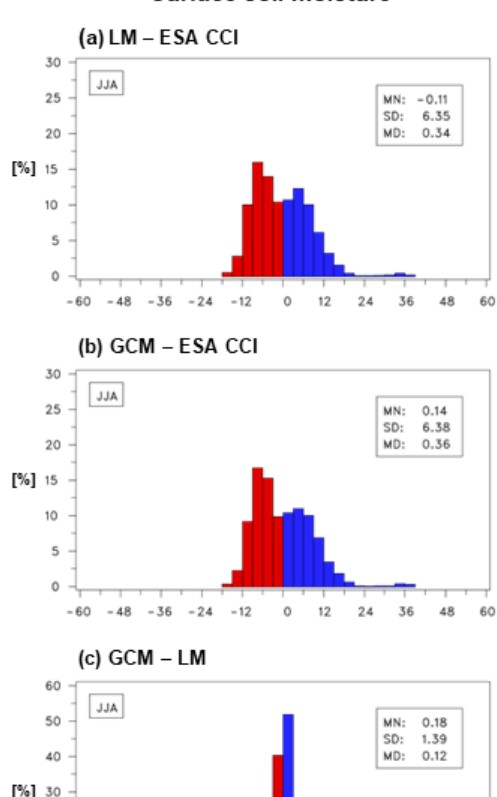

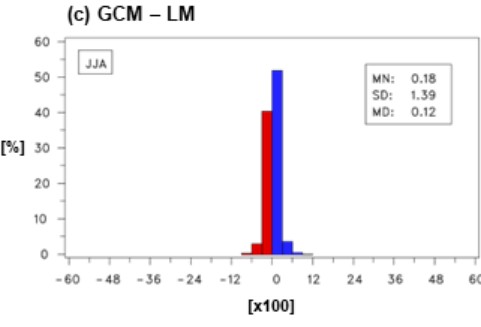

**Figure 4:** Distributions of the area-weighted seasonal mean differences of surface soil moisture (a) between LM and ESA CCI (LM – ESA CCI), (b) between GCM and ESA CCI (GCM – ESA CCI) and (c) between GCM and LM. Units are $m^3/m^3$ (x100) on the abscissa and % on the ordinate, respectively. The mean, standard deviation and median of the distribution are given in the box; units are $m^3/m^3$ (x100). Note the swap of the colour scale with red colours indicating negative differences and blue colours positive ones as well as the different scale on the ordinate in (c).





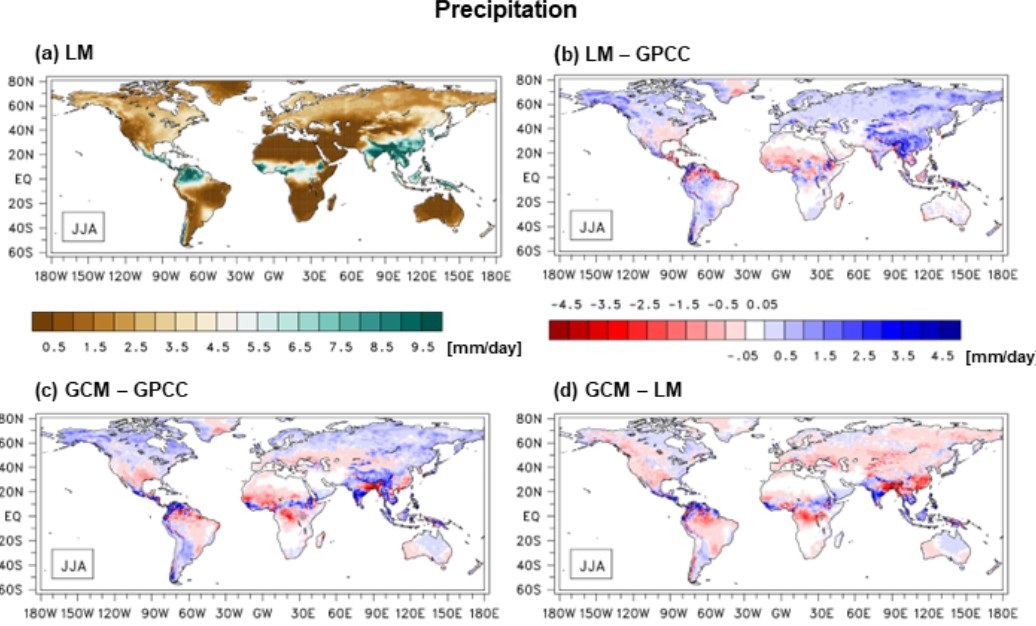

**Figure 5:** Seasonal mean values of daily precipitation for 1979-2017 for LM (a) as well as the differences (b) between LM and GPCC (LM – GPCC), (c) between GCM and GPCC (GCM – GPCC) and (d) between GCM and LM. Units are mm/day; the contour interval is 0.5 mm/day in (a) and 0.5 mm/day in (b-d), respectively. Note the swap of the colour scale with red colours indicating small values (a) or negative differences (b-d) and blue colours indicating high values (a) or positive differences (b-d).










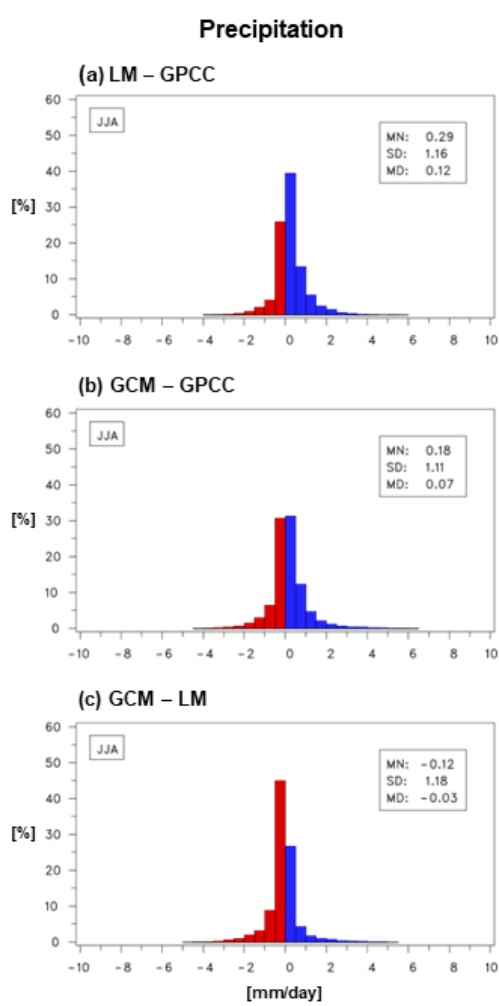

**Figure 6:** Distributions of the area-weighted seasonal mean differences of daily precipitation (a) between LM and GPCC (LM – GPCC), (b) between GCM and GPCC (GCM – GPCC) and (c) between GCM and LM. Units are mm/day on the abscissa and % on the ordinate, respectively. The mean, standard deviation and median of the distribution are given in the box; units are mm/day. Note the swap of the colour scale with red colours indicating negative differences and blue colours positive ones.

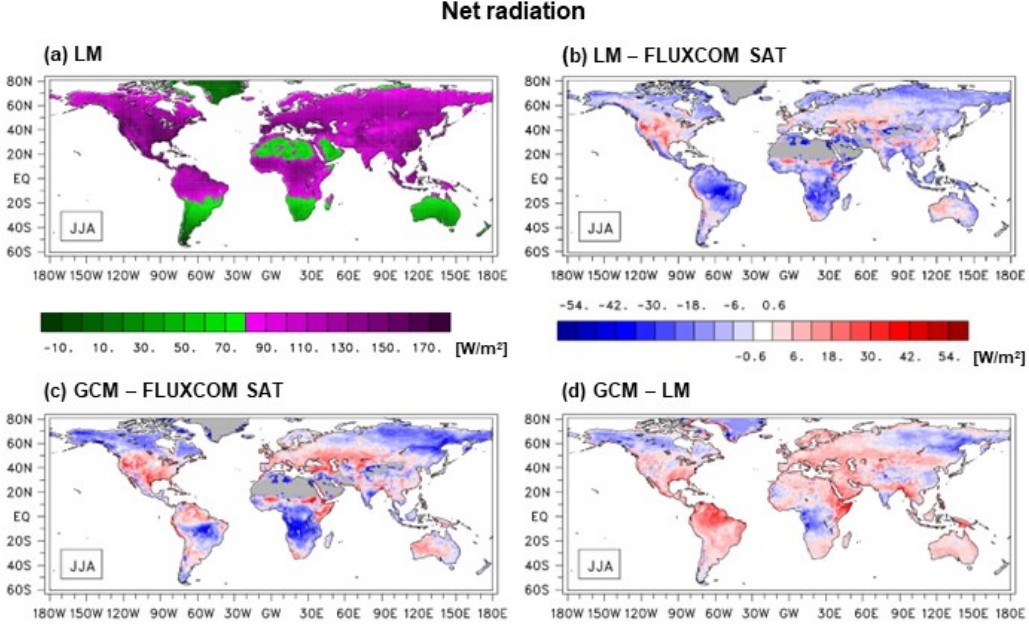

**Figure 7:** Seasonal mean values of net radiation (incoming) for 2001-2015 for LM (a) as well as the differences (b) between LM and FLUXCOM SAT (LM – FLUXCOM SAT), (c) between GCM and FLUXCOM SAT (GCM – FLUXCOM SAT) and (d) between GCM and LM. Units are W/m$^2$; the contour interval is 10 W/m$^2$ in (a) and 6 W/m$^2$ (b-d), respectively. Missing data in FLUXCOM SAT are marked in grey.










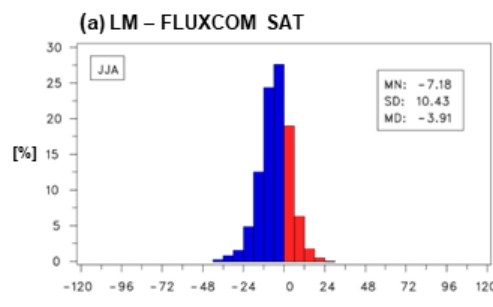

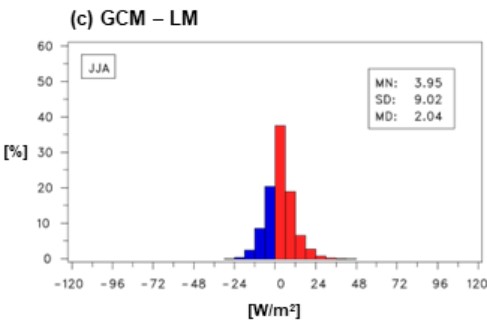


**Figure 8:** Distributions of the area-weighted seasonal mean differences of net radiation (a) between LM and FLUXCOM SAT (LM – FLUXCOM SAT), (b) between GCM and FLUXCOM SAT (GCM – FLUXCOM SAT) and (c) between GCM and LM. Units are W/m² on the abscissa and % on the ordinate, respectively. The mean, standard deviation and median of the distribution are given in the box; units are W/m². Note the different scale on the ordinate in (c).








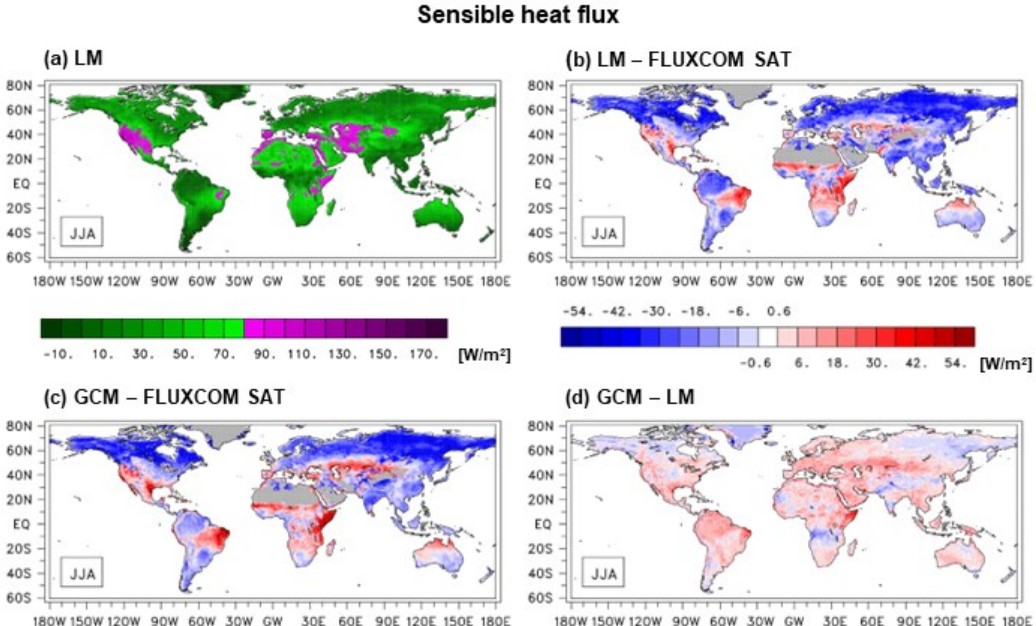

**Figure 9:** As Fig. 7 but for sensible heat flux.








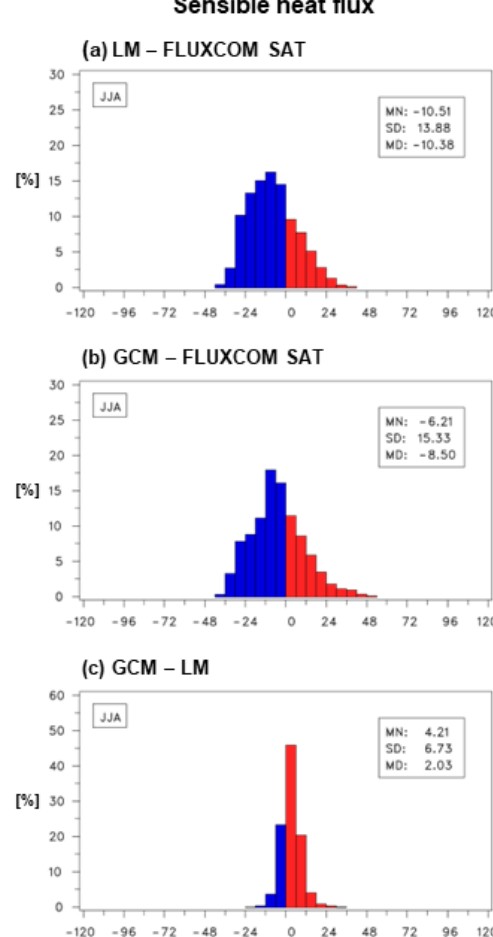

**Figure 10:** As Fig. 8 but for sensible heat flux. Note the different scale on the ordinate in (c).






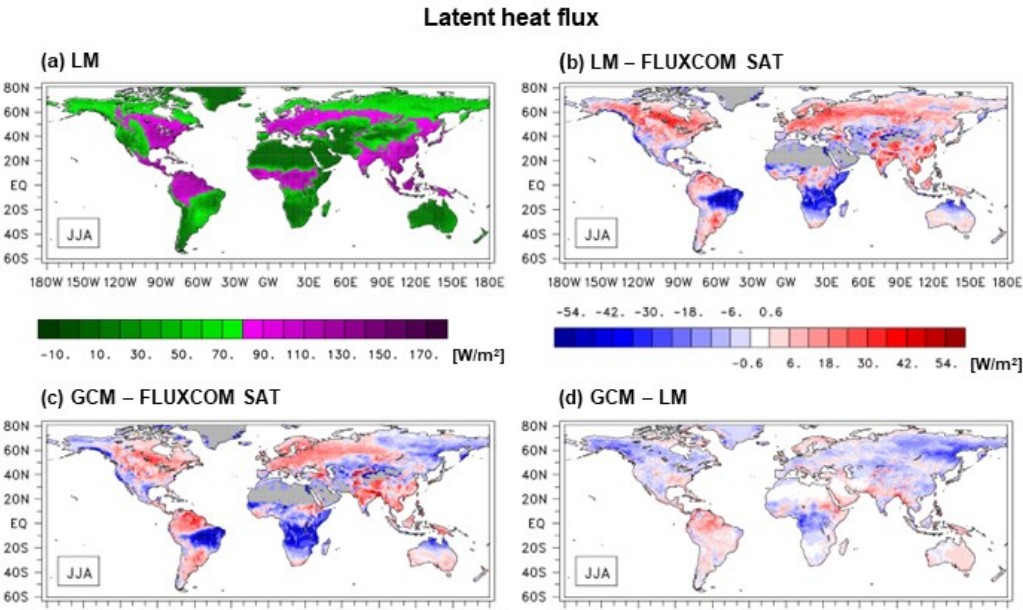

**Figure 11:** As Fig. 7 but for latent heat flux.










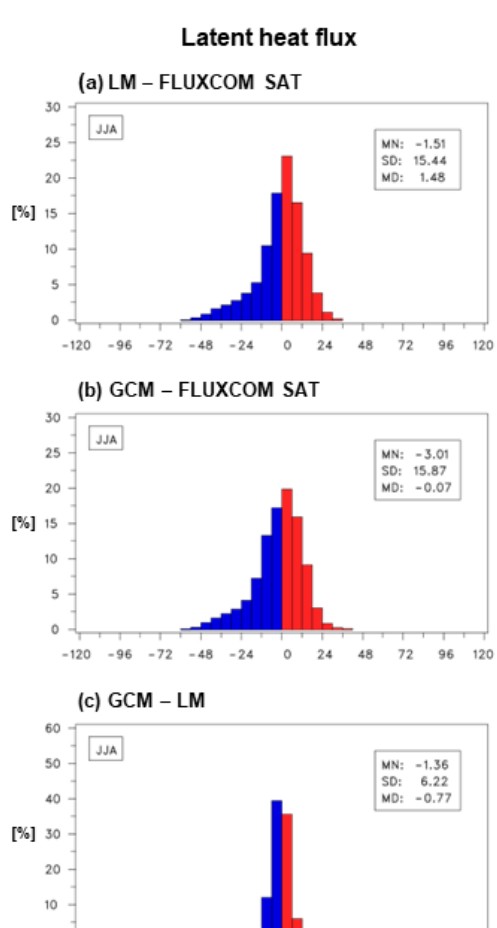

**Figure 12:** As Fig. 8 but for latent heat flux. Note the different scale on the ordinate in (c).




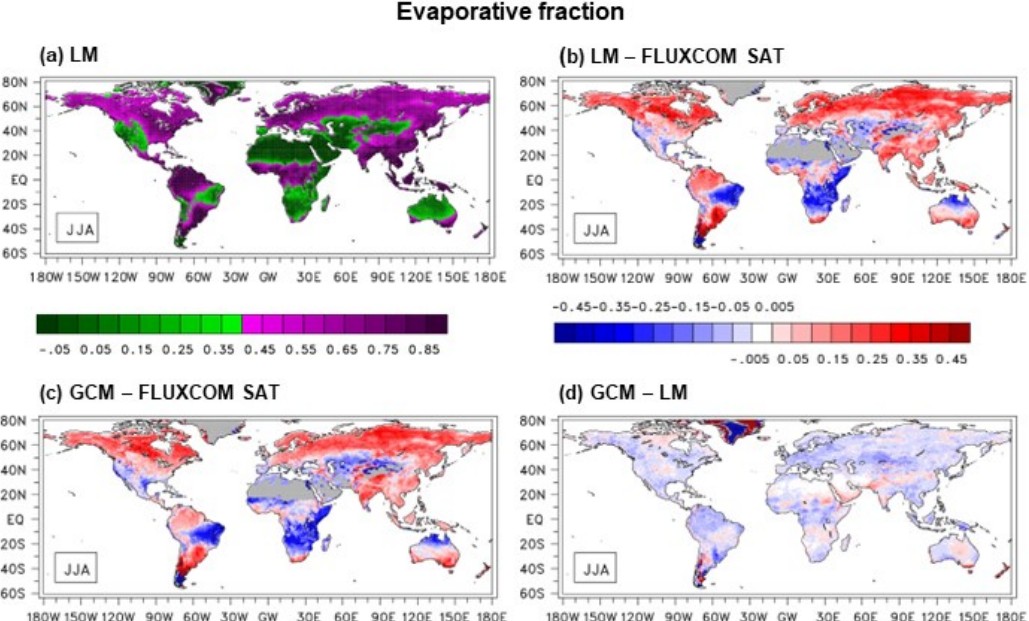

**Figure 13:** Seasonal mean values of the evaporative fraction for 2001-2015 for LM (a) as well as the differences (b) between LM and FLUXCOM SAT (LM – FLUXCOM SAT), (c) between GCM and FLUXCOM SAT (GCM – FLUXCOM SAT) and (d) between GCM and LM. Units are standard unit; the contour interval is 0.05 in (a) and (b-d), respectively. Missing data in FLUXCOM SAT are marked in grey.










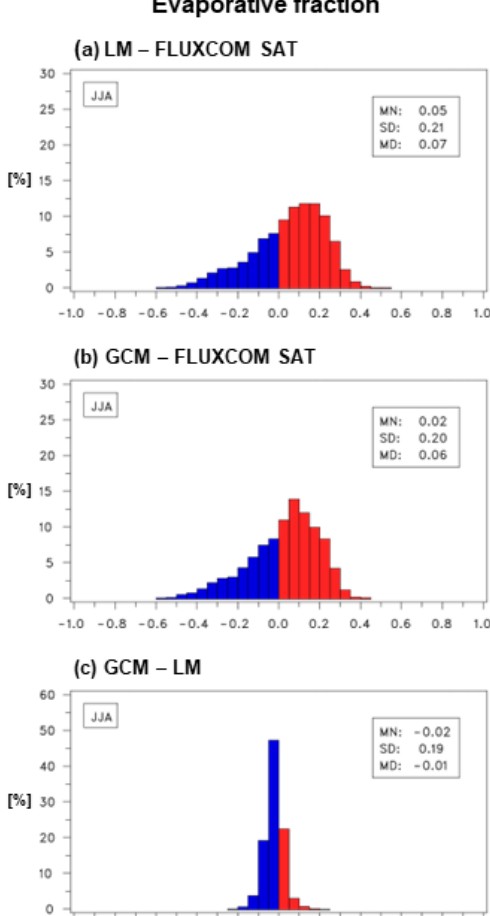

**Figure 14:** Distributions of the area-weighted seasonal mean differences of the evaporative fraction (a) between LM and FLUXCOM SAT (LM – FLUXCOM SAT), (b) between GCM and FLUXCOM SAT (GCM – FLUXCOM SAT) and (c) between GCM and LM. Units are standard unit on the abscissa and % on the ordinate, respectively. The mean, standard deviation and median of the distribution are given in the box; units are standard unit. Note the different scale on the ordinate in (c). Areas with missing data in the respective reference data sets are marked in grey.


## Surface energy fluxes

### a) LM – FLUXCOM SAT / ERA5 / ESA CCI

### b) GCM – FLUXCOM SAT / ERA5 / ESA CCI

**Figure 15:** Differences in the regional seasonal means of surface energy fluxes, i.e., net radiative flux (RAD), fluxes of sensible (SHF) and latent heat (LHF) and the evaporative fraction (EF), as well as land-surface temperature (LST) and surface soil moisture (SM) (a) between LM and the reference data sets and (b) between GCM and the reference data sets, i.e., ERA5 for LST, ESA CCI for SM and FLUXCOM SAT for the surface energy fluxes for 2003-2014. Units and contour intervals can be seen in the corresponding bars.



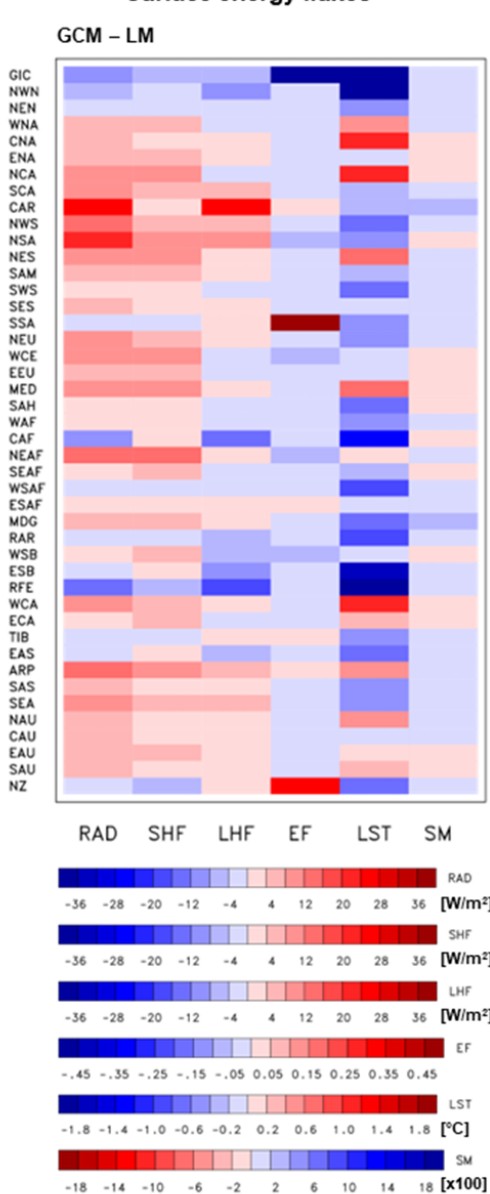


**Figure 16:** Differences in the regional seasonal means of surface energy fluxes, i.e., net radiative flux (RAD), fluxes of sensible (SHF) and latent heat (LHF) and the evaporative fraction (EF), as well as land-surface temperature (LST) and surface soil moisture (SM) between LM GCM for 2003-2014. Units and contour intervals can be seen in the corresponding bars.




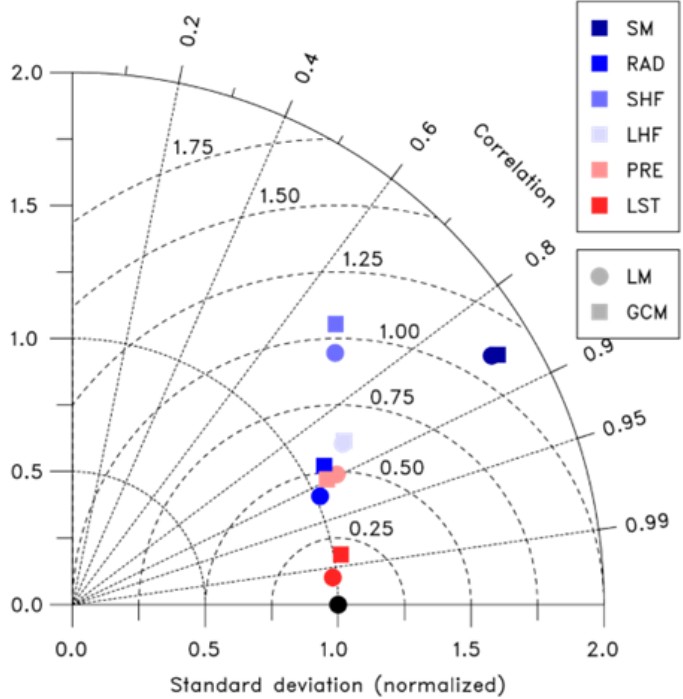


**Figure 17:** Taylor diagram for the seasonal means of land-surface temperature (LST), the surface energy fluxes, i.e., net radiative flux (RAD), fluxes of sensible (SHF) and latent heat (LHF), surface soil moisture (SM) and precipitation (PRE) for LM (filled circles) and GCM (filled squares) with respect to the different reference data, i.e., ERA5 for LST, FLUXCOM SAT for the surface energy fluxes, ESA-CCI for SM and GPCC for PRE. Periods considered are 1979-2017 for LST, SM and PRE and 2001-2015 for RAD, SHF and LHF, respectively.

All values, i.e., the root-mean-square difference and standard deviations, are normalized by the standard deviations of the corresponding reference data.