# Peer review of "The role of land-surface interactions for surface climate in the EC-Earth3 earth system model"

_Earth System Dynamics, 2023_

## Referee Comment (RC1)

MS Title: The role of land-surface interactions for surface climate in the EC-Earth3 earth system model

Author: Wilhelm May

**General comment**

The manuscript entitled "The role of land-surface interactions for surface climate in the EC-Earth3 earth system model" by Wilhelm May evaluates two simulations against multiple reference data sets. For the first simulation, EC-Earth3 is forced offline with reanalysis data. For the second simulation, the land surface and atmosphere are dynamically coupled, but land surface characteristics are prescribed from the offline run. Comparing offline and constrained online simulations shows to what extent biases are related to the land surface component versus the coupling of the atmosphere. The first simulation has an overall warm and dry bias with some regional differences, a negative bias in net radiation and sensible heat flux, and a positive bias for latent heat flux. Dynamic coupling leads to stronger biases, and the sign of those biases can differ from the ones in the offline run. The study describes regional differences using multiple reference data sets in great detail. The author concludes that atmospheric coupling has a large impact on temperature, net radiation and sensible heat flux, and a modest impact on soil moisture and latent heat flux. The study concludes that further model development is required for reducing model biases.

The manuscript is very well written and I expect that the results are of much value to EC-Earth3 model developers. However, the manuscript also remains very descriptive. The Discussion section elaborates only on the study's limitations. It remains unclear how well the results compare to previous studies, the relevance of the findings for the broader land surface modelling community, and possible venues for model improvements. I, therefore, recommend that the manuscript may be considered for publication in ESD after major revision. Please find my detailed comments below.

**Detailed Comments**

L 7 The effect of the land surface on the *global* climate system is not limited to $CO_2$ alone. The ice and snow albedo feedback for instance affects the global climate as well. Please rewrite.

L 65 The description of how changes in latent heat flux can potentially affect precipitation is vague, please rewrite.

L 76 Please add references.

L 88 Typo, replace *ad* with *and*.

L 176 Explain why you use prescribed data from the offline simulation in your coupled simulation.

L 222 So your offline simulation is based on a spinup that was driven with GSWP3 and a historical run that uses ERA5 data? Why did you not use ERA5 data for both, the historical and the the spinup? Please justify.

L 246 Replace *merged* with *spatially upscaled*.

L 297 Please gather all results under a *Results* section.

L 307 Please add the value of the global mean bias in the text.

L 370 The equation applies to the upwelling, not downwellig LW radiation. Please adjust the direction of the arrow accordingly.

L 371 Add that ground heat flux is close to zero when averaged over longer periods, such as a month, which is why you may omit it in your equation.

L 358-377 This paragraph provides very basic information that you can expect the reader to know. I think you can skip this part.

L 402 Provide the definition of evaporative fraction further up where you first use the term.

L 587 This paragraph explains why you focus on JJA. Please provide this justification earlier in the methods section.

L 570 The Discussion section only elaborates on the study's limitation. Please also elaborate on how well your results compare to previous studies, the relevance of your findings for the wider land surface modeling community, and possible venues for model improvements.

L 605 I would omit subsections in the Summary and conclusions section

Code and Data availability: You write that code and data can be made available by the author on request. I don't think this approach satisfies the ESD data policy, which states that data and other information underpinning the research findings are findable, accessible, interoperable, and reusable (FAIR) not only for humans but also for machines.

**Figures**

- Figures 1a, 3a, 5a, 7a, and 9a: Please replace the divergent with a continuous color legend, as you are not showing the difference but the absolute values. Also, please ensure all legends include their units.

- Figures that show seasonal biases: Please denote where biases are statistically significant.

- Figure 15: Please add a map that identifies the different IPCC regions.

- The text includes 17 Figures, of which not all are discussed in great detail. I recommend to remove figures that are not discussed in more detail, such that the total number of figures does not exceed ten.

---

## Referee Comment (RC2)

**Summary**

This manuscript evaluates boreal summer season mean surface fluxes, soil moisture, land surface temperature (LST) and precipitation in two EC-Earth3 earth system model (ESM) simulations – one using the land-surface component only and one coupled to the atmospheric component. The objective of the work is to assess the quality of the land-surface component and to determine the impact of coupling on the simulation of surface climate.

**Review**

In order to make the best use of ESMs for climate research, it is critical to understand how well they simulate the real Earth climate. The current work evaluates EC-Earth3 and the results would be informative for scientists looking to use that model to answer research questions. In particular the current work focuses on the impact of using the coupled configuration versus the land-only configuration on biases in the simulation of surface climate. This is an important consideration, as it is necessary to understand whether biases in the land-surface model propagate through the coupled model and how they may be enlarged or supressed by the model representation of land-atmosphere interactions.

However for a paper with the title "the role of land-surface interactions for surface climate in the EC-Earth3 earth system model" and the aim of evaluating biases globally I find the usefulness of the current results limited due to the sole focus on boreal summer and the slim selection of observational datasets used for evaluating LST and precipitation.

Land-atmosphere coupling in the southern hemisphere is maximised in boreal winter, and I suspect the biases in surface climate will be also. An analysis for multiple seasons (at least June-August for the Northern hemisphere and December-February for the Southern hemisphere) is required. The analysis would also benefit from the inclusion of additional observational datasets including monthly mean satellite LST (infrared and microwave for comparison) and at least one additional precipitation dataset.

I would therefore recommend that the current manuscript be revised before being considered for publication. Please find my specific comments below.

**Specific Comments**

*Focus on boreal summer*

Line 300 is the first line in the main text that mentions the study focuses only on boreal summer (June – August). The study focus should be reflected in the abstract, introduction and possibly the title, with the justification provided at the start of the manuscript, not in the Discussion.

On that note, I am not convinced by the reasoning provided on Lines 587-593. If the objective is to understand the effect of atmospheric coupling on surface climate biases globally the work should consider the different seasons when the coupling is maximised in the different regions.

*Datasets – LST*

The author does not provide a justification for the exclusion of satellite LST products in the data section. Instead the author includes an inaccurate statement in the results section:

"A serious shortcoming of existing satellite-based data sets for land-surface temperature is that they depend on clear-sky conditions and, thus, do not incorporate periods with partly or fully cloudy conditions".

This is actually only true for infrared satellite observations. Microwave satellite observations (for example AMSR2 skin temperature) do not depend on clear sky conditions. Furthermore the limitation of infrared satellite observations does not prevent a comparison when considering seasonal mean values. You could apply a mask to the MODIS Terra/Aqua monthly mean daytime LST products to exclude land points with less than a minimum number of valid observations (clear sky days) contributing to the value (for example 10 days in the season).

Therefore it is possible to include satellite LST products in the evaluation rather than rely solely on reanalysis.

One final query on this point, why did the authors use LST from ERA5 rather than ERA5-Land?

AMSR2/GCOM-W1 surface soil moisture (LPRM) L3 1 day 10 km x 10 km ascending V001 (contains microwave skin temperature)
DOI:10.5067/B0GHODHJLDA8

MODIS/Aqua Monthly mean Day-Time Land Surface Temperature at 1x1 degree V005
DOI:10.5067/2YCD3NSNDMRM

MODIS/Terra Monthly mean Day-Time Land Surface Temperature at 1x1 degree V005
DOI:10.5067/4SI45J6G6BW5

*Datasets – precipitation*

Were other datasets aside from GPCC considered? For example CHIRPS?

It would be more informative if more than one dataset were included for comparison.

The author needs to justify their choice of datasets in the data section.

*Abstract*

The abstract could be simplified to make it more clear and concise for the reader. For example I find this line from the Discussion makes the aims of the work immediately apparent:

"the intention is to assess a) the quality of the land-surface component and b) the effects of the coupling with the atmosphere"

Compared to the equivalent line from the abstract which is much longer and includes model details that could be reserved for later:

"The aim of this study is twofold, first to evaluate the quality of the simulation of surface climate by the land-surface component of the EC-Earth3 ESM, combining the HTESSEL land-surface model and the LPJ-GUESS dynamic vegetation model, and second to assess the role of the coupling of the land surface with the atmosphere for the simulation of the surface climate in EC-Earth3."

Minor point on Line 21-22, I assume one of the instances of "underestimate" should in fact read "overestimate"?

*Introduction*

In paragraph 2 the author states that EC-Earth3 represents a distinct step forwards, then proceeds to list the flaws with EC-Earth3. How do the biases in EC-Earth3 represent an improvement compared to the earlier version of EC-Earth?

There is a distinct lack of references in paragraphs 3-5 although many different processes are discussed. Here are a few examples that could be included in paragraph 3:

K. L. Findell, E. A. B. Eltahir, Atmospheric controls on soil moisture-boundary layer interactions. Part I: Framework development. J. Hydrometeorol. 4, 552–569 (2003)

R. A. Pielke, Influence of the spatial distribution of vegetation and soils on the prediction of cumulus convective rainfall. Rev. Geophys. 39, 151–177 (2001)

Bhowmick, M. and Parker, D.J. (2018) Analytical solution to a thermodynamic model for the sensitivity of afternoon deep convective initiation to the surface Bowen ratio. *Quarterly Journal of the Royal Meteorological Society*, **144**(716), 2216– 2229. https://doi.org/10.1002/qj.3340.

Concerning paragraph 3, the concepts are a little disorganised. The paragraph is predominately discussing the surface impact on the atmosphere, yet ends with a sentence on

the atmospheric impact on the surface. Also the discussion on surface impacts on the atmosphere jumps between temperature and humidity/precipitation couplings rather than discussing each in turn.

*Equations*

The flux equations, currently in the results section, should be included in the data section.

---

## Author Comment (AC1)

**Reply to Reviewer 1, August 2023**

Thank you for the detailed review and the suggestions to improve the manuscript! In the following I will respond to the comments in blue. Not all responses are full-fledged, but rather indicate how to implement the required changes in the revised manuscript. I will start revising the manuscript accordingly after the editor's decision to proceed further.

**General Comment**

The manuscript entitled "The role of land-surface interactions for surface climate in the EC-Earth3 earth system model" by Wilhelm May evaluates two simulations against multiple reference data sets. For the first simulation, EC-Earth3 is forced offline with reanalysis data. For the second simulation, the land surface and atmosphere are dynamically coupled, but land surface characteristics are prescribed from the offline run. Comparing offline and constrained online simulations shows to what extent biases are related to the land surface component versus the coupling of the atmosphere. The first simulation has an overall warm and dry bias with some regional differences, a negative bias in net radiation and sensible heat flux, and a positive bias for latent heat flux. Dynamic coupling leads to stronger biases, and the sign of those biases can differ from the ones in the offline run. The study describes regional differences using multiple reference data sets in great detail. The author concludes that atmospheric coupling has a large impact on temperature, net radiation and sensible heat flux, and a modest impact on soil moisture and latent heat flux. The study concludes that further model development is required for reducing model biases.

The manuscript is very well written and I expect that the results are of much value to EC-Earth3 model developers. However, the manuscript also remains very descriptive. The Discussion section elaborates only on the study's limitations. It remains unclear how well the results compare to previous studies, the relevance of the findings for the broader land surface modelling community, and possible venues for model improvements. I, therefore, recommend that the manuscript may be considered for publication in ESD after major revision. Please find my detailed comments below.

I acknowledge the main concern that the study is very EC-Earth specific and that the results need to be discussed in relation to the aspects mentioned here. I am not sure about previous studies that have followed this approach, but I will check the scientific literature once more. Maybe I can learn something from overview papers on the models used for CMIP. I think I can include a discussion of the relevance of the results for both land-surface and atmospheric modelling and speculate about how the finding can support model development.

**Detailed Comments**

L 7 The effect of the land surface on the global climate system is not limited to $CO_2$ alone. The ice and snow albedo feedback for instance affects the global climate as well. Please rewrite.

Yes. I will also check the Introduction for completeness.

L 65 The description of how changes in latent heat flux can potentially affect precipitation is vague, please rewrite.

Yes. My intention was to be short here. Will add more details.

L 76 Please add references.

Yes.

L 88 Typo, replace *ad* with *and*.

Correct.

L 176 Explain why you use prescribed data from the offline simulation in your coupled simulation.

In short, that is for eliminating the contributions of errors in the representation of the land surface to the biases induced by errors in the atmospheric model.

L 222 So your offline simulation is based on a spinup that was driven with GSWP3 and a historical run that uses ERA5 data? Why did you not use ERA5 data for both, the historical and the spinup? Please justify.

At the time when I performed the offline simulation, ERA5 was not yet available before 1979. That is why I had to use a "realistic" initial state of the land-surface component, particularly in the case of LPJ-GUESS, from an offline simulation with a meteorological forcing that went back earlier in time to begin with. To avoid the impacts of discrepancies between ERA5 and GSWP3, I first ran the offline model for 20 years forced with the climatological meteorological forcing from ERA5 ahead of the offline simulation with the varying forcing. I can add this reasoning to the text, but I am not sure how detailed it should be.

L 246 Replace *merged* with *spatially upscaled*.

Fine.

L 297 Please gather all results under a Results section.

I guess you refer to Figure 17 here, which I had added (in the Results section, actually) because I think the Taylor diagram is well suited to summarize (some of) the results of the study in a comprehensive way. I am not sure how to deal with the high number of figures (see also my response below), but this figure could be a candidate for omission.

L 307 Please add the value of the global mean bias in the text.

Yes, this improves the readability. Will also add other values from Table 2 to the text where relevant.

L 370 The equation applies to the upwelling, not downwelling LW radiation. Please adjust the direction of the arrow accordingly.

Yes, that is right, but probably not necessary anymore when omitting this part.

L 371 Add that ground heat flux is close to zero when averaged over longer periods, such as a month, which is why you may omit it in your equation.

Yes, but probably not necessary anymore when omitting this part.

L 358-377 This paragraph provides very basic information that you can expect the reader to know. I think you can skip this part.

I agree. I could mention the components that are included here (net radiation, sensible and latent heat flux, evaporative fraction) in Section 3.1 add that the relationship between net radiation and the energy fluxes.

L 402 Provide the definition of evaporative fraction further up where you first use the term.

Yes, I think I will add this to Section 3.1.

L 587 This paragraph explains why you focus on JJA. Please provide this justification earlier in the methods section.

Yes, I should have done that somewhere in the beginning. The other reviewer suggests that I consider the DJF season for the Southern Hemisphere, as the coupling between the land surface and the atmosphere there is stronger than for JJA. This justification would hold for both hemispheres.

L 570 The Discussion section only elaborates on the study's limitation. Please also elaborate on how well your results compare to previous studies, the relevance of your findings for the wider land surface modeling community, and possible venues for model improvements.

Yes, I can see that the study is very EC-Earth specific and that the results need to be discussed in relation to the aspects mentioned here. I am not sure about previous studies that have followed this approach, but I will check the scientific literature once more. Maybe I can learn something from overview papers on the models used for CMIP. I think I can include a discussion of the relevance of the results for both land-surface and atmospheric modelling and speculate about how the finding can support model development.

L 605 I would omit subsections in the Summary and conclusions section.

Yes, I can see this looks a bit awkward, given the shortness of the sub-sections.

Code and Data availability: You write that code and data can be made available by the author on request. I don't think this approach satisfies the ESD data policy, which states that data and other information underpinning the research findings are findable, accessible, interoperable, and reusable (FAIR) not only for humans but also for machines.

I need to check up on this. I had just looked at what had been done in other publications in Erath System Dynamics, where code and data were made available on request in some of them. The code (cdo and Fortan) used for the analysis, I think, is standard and it might not be necessary to make it available through a platform like github. The (monthly mean) data from the simulations, on the other hand, I might need to make available be on a suitable platform.

**Figures**

• Figures 1a, 3a, 5a, 7a, and 9a: Please replace the divergent with a continuous color legend, as you are not showing the difference but the absolute values. Also, please ensure all legends include their units.

I presume, this comment includes Figures 11a and 13a as well. Yes, I can see the point (not so much for soil moisture and precipitation, though). But for the energy fluxes there are slightly negative values, which need a be treated somewhat differently. For the latter, I presume you refer to the evaporative fraction. In this case, the unit is standard unit, which I did not add to the legend.

• Figures that show seasonal biases: Please denote where biases are statistically significant.

I had a version of the maps indicating the statistical significance according to a Student's t-test. Given the size of the maps, I thought it was a bot hard to see. I will give it a try again (indicating either the significant or non-significant areas) and see how well it works. Alternatively, I could estimate the area of significant differences and add the numbers to table 2.

• Figure 15: Please add a map that identifies the different IPCC regions.

I could add the respective map from Iturbide et al. (2020) to the supplementary material, complementing Table S1.

• The text includes 17 Figures, of which not all are discussed in great detail. I recommend to remove figures that are not discussed in more detail, such that the total number of figures does not exceed ten.

I could omit Figure 17 (see above), and it might also be possible to omit Figures 15 and 16. The latter, because of the other reviewer's suggestion to consider the JJA season for the Northern and the DJF season for the Southern Hemisphere. If I do that, then Table S1 and Figure S12 will also be removed from the supplementary material.

Another way to reduce the number of figures could be to combine the maps with the histograms in one figure for each variable. Possibly (some of) the histograms could be omitted, where they are not really discussed. The maps, I think, I need to keep. In the same way, the number of figures in the supplementary material could be reduced, possibly removing the histograms.

Possibly, I could also omit Figure S3 from the supplementary material because of the small differences between the two versions of the GLEAM data and just mention it in the text.

---

## Author Comment (AC2)

**Reply to Reviewer 2, August 2023**

Thank you for the thorough review and the suggestions to improve the manuscript! In the following I will respond to the comments in blue. Not all responses are full-fledged, but rather indicate how to implement the required changes in the revised manuscript. I will start revising the manuscript accordingly after the editor's decision to proceed further.

**Summary**

This manuscript evaluates boreal summer season mean surface fluxes, soil moisture, land surface temperature (LST) and precipitation in two EC-Earth3 earth system model (ESM) simulations – one using the land-surface component only and one coupled to the atmospheric component. The objective of the work is to assess the quality of the land-surface component and to determine the impact of coupling on the simulation of surface climate.

**Review**

In order to make the best use of ESMs for climate research, it is critical to understand how well they simulate the real Earth climate. The current work evaluates EC-Earth3 and the results would be informative for scientists looking to use that model to answer research questions. In particular the current work focuses on the impact of using the coupled configuration versus the land-only configuration on biases in the simulation of surface climate. This is an important consideration, as it is necessary to understand whether biases in the land-surface model propagate through the coupled model and how they may be enlarged or supressed by the model representation of land-atmosphere interactions.

However for a paper with the title "the role of land-surface interactions for surface climate in the EC-Earth3 earth system model" and the aim of evaluating biases globally I find the usefulness of the current results limited due to the sole focus on boreal summer and the slim selection of observational datasets used for evaluating LST and precipitation.

Land-atmosphere coupling in the southern hemisphere is maximised in boreal winter, and I suspect the biases in surface climate will be also. An analysis for multiple seasons (at least June-August for the Northern hemisphere and December-February for the Southern hemisphere) is required. The analysis would also benefit from the inclusion of additional observational datasets including monthly mean satellite LST (infrared and microwave for comparison) and at least one additional precipitation dataset.

Yes, I think the observation on the choice of the season is correct. I had chosen to present only season to limit the number of figures (which already is quite large) and motivated the focus on JJA in the text with the domination of the Northern and Southern Hemisphere. I will extend the analysis to also include DJF, focussing on the Northern Hemisphere (10 °S – 80 °N) in JJA and the Southern Hemisphere (60 °S – 10 °N) in DJF. Please find more details below.

I think it is correct that observations cannot be a "perfect" representation of the true climate state and that, therefore, several independent observational data sets should be considered. For the land-surface temperature, we face the problem that may of the satellite-derived estimates consider only retrievals under clear-sky conditions, limiting the choices. For

precipitation, there are several options based on gauge data, satellite retrievals or combinations thereof, but in this case the spatial scale of the data must be taken into account to avoid differences that are due to the different representations of orography or coast lines. I will include an additional data set for land surface temperature and precipitation. Please find more details below.

I would therefore recommend that the current manuscript be revised before being considered for publication. Please find my specific comments below.

**Specific Comments**

*Focus on boreal summer*

Line 300 is the first line in the main text that mentions the study focuses only on boreal summer (June – August). The study focus should be reflected in the abstract, introduction and possibly the title, with the justification provided at the start of the manuscript, not in the Discussion.

Yes, I agree. The choice of the JJA season should be more obvious throughout the manuscript.

On that note, I am not convinced by the reasoning provided on Lines 587-593. If the objective is to understand the effect of atmospheric coupling on surface climate biases globally the work should consider the different seasons when the coupling is maximised in the different regions.

As mentioned above, I will extend the analysis to also include DJF, focussing on the Northern Hemisphere (10 °S – 80 °N) in JJA and the Southern Hemisphere (60 °S – 10 °N) in DJF. Not presenting the whole globe (60 °S – 80 °N) in the maps is necessary for not increasing the number of figures. At the same time, the maps need to be combined with the histograms, unless the histograms can be omitted as such. Moreover, I consider omitting the figures with the averages for the IPCC-regions (Figs. 15, 16 & S12) and Table S1, because the regions are located in different hemispheres and, thus, represent different seasons.

*Datasets – LST*

The author does not provide a justification for the exclusion of satellite LST products in the data section. Instead the author includes an inaccurate statement in the results section:

"A serious shortcoming of existing satellite-based data sets for land-surface temperature is that they depend on clear-sky conditions and, thus, do not incorporate periods with partly or fully cloudy conditions".

This is actually only true for infrared satellite observations. Microwave satellite observations (for example AMSR2 skin temperature) do not depend on clear sky conditions. Furthermore the limitation of infrared satellite observations does not prevent a comparison when considering seasonal mean values. You could apply a mask to the MODIS Terra/Aqua monthly mean daytime LST products to exclude land points with less than a minimum number of valid observations (clear sky days) contributing to the value (for example 10 days in the season).

Therefore it is possible to include satellite LST products in the evaluation rather than rely solely on reanalysis.

One final query on this point, why did the authors use LST from ERA5 rather than ERA5-Land?

That is because ERA5 has the lower resolution, which is closer to the resolution of the model. Using the high-resolution data from ERA5-Land would introduce some artificial biases in areas with high and steep different orography. See, for example, Figure 4 in Muñoz-Sabatier et al. (2021).

AMSR2/GCOM-W1 surface soil moisture (LPRM) L3 1 day 10 km x 10 km ascending V001 (contains microwave skin temperature) DOI:10.5067/B0GHODHJLDA8

MODIS/Aqua Monthly mean Day-Time Land Surface Temperature at 1x1 degree V005 DOI:10.5067/2YCD3NSNDMRM

MODIS/Terra Monthly mean Day-Time Land Surface Temperature at 1x1 degree V005 DOI:10.5067/4SI45J6G6BW5

The two doi's point to the monthly mean day-time LST at a resolution of 1° from the Aqua and the Terra satellites, respectively. Data from Terra are available for the period March 2000 to June 2015 and data from Aqua for the period August 2002 to June 2015, resulting in only 12 years of data. But there are also the corresponding data sets for the night-time LST. Although combing the four data sets seems to overcome much of the issue with the clear-sky conditions (Chen et al. 2018), Muñoz-Sabatier et al. (2021) noticed remaining problems in regions with permanent cloud cover sauch as the tropical rain forests. I will include the combined MODIS LST data in the study, having the limitations in mind when interpreting the differences.

*Datasets – precipitation*

Were other datasets aside from GPCC considered? For example CHIRPS?

It would be more informative if more than one dataset were included for comparison.

The author needs to justify their choice of datasets in the data section.

https://www.chc.ucsb.edu/data/chirps

I am aware that different data sets of precipitation can give somewhat different estimates, even for long-term climatologies, given the differences in how these data sets are assembled. For the purpose of this study, I hadn't considered using another precipitation dataset, mainly because I did not consider the precipitation bias not very essential for the study. The main intention was to indicate a potential bias in the precipitation in ERA5, which had been used as forcing in the offline simulation.

Unfortunately, the CHIRPS "global" data (starting in January 1981) do not cover the entire globe but only the land areas between 50 °S and 50 °N. Also, they are at a very high resolution of 0.05°. The GPCP monthly data (starting in January 1979), the other hand, are at a rather low resolution of 2.5°, whereas the high-resolution (0.5 °) GPCP daily data do not start before June 2000. I will try different options and include the best suited in the manuscript.

*Abstract*

The abstract could be simplified to make it more clear and concise for the reader. For example I find this line from the Discussion makes the aims of the work immediately apparent:

"the intention is to assess a) the quality of the land-surface component and b) the effects of the coupling with the atmosphere"

Compared to the equivalent line from the abstract which is much longer and includes model details that could be reserved for later:

"The aim of this study is twofold, first to evaluate the quality of the simulation of surface climate by the land-surface component of the EC-Earth3 ESM, combining the HTESSEL land-surface model and the LPJ-GUESS dynamic vegetation model, and second to assess the role of the coupling of the land surface with the atmosphere for the simulation of the surface climate in EC-Earth3."

Yes. I agree. I will rephrase this part accordingly.

Minor point on Line 21-22, I assume one of the instances of "underestimate" should in fact read "overestimate"?

Yes, this is correct.

*Introduction*

In paragraph 2 the author states that EC-Earth3 represents a distinct step forward, then proceeds to list the flaws with EC-Earth3. How do the biases in EC-Earth3 represent an improvement compared to the earlier version of EC-Earth?

Yes, I realize this is somewhat contradictory. I will consult Döscher et al. (2022) for details on the improvement compared to the previous model version.

There is a distinct lack of references in paragraphs 3-5 although many different processes are discussed. Here are a few examples that could be included in paragraph 3:

K. L. Findell, E. A. B. Eltahir, Atmospheric controls on soil moisture-boundary layer interactions. Part I: Framework development. J. Hydrometeorol. 4, 552–569 (2003)

R. A. Pielke, Influence of the spatial distribution of vegetation and soils on the prediction of cumulus convective rainfall. Rev. Geophys. 39, 151–177 (2001)

Bhowmick, M. and Parker, D.J. (2018) Analytical solution to a thermodynamic model for the sensitivity of afternoon deep convective initiation to the surface Bowen ratio. Quarterly Journal of the Royal Meteorological Society, 144, 2216–2229.

Thank You for pointing this out and providing some references. I will include additional references in relation to the different processes mentioned.

Concerning paragraph 3, the concepts are a little disorganised. The paragraph is predominately discussing the surface impact on the atmosphere, yet ends with a sentence on the atmospheric impact on the surface. Also the discussion on surface impacts on the

atmosphere jumps between temperature and humidity/precipitation couplings rather than discussing each in turn.

*I tried to keep this part concise, apparently this means that the line of thought got disturbed. I will rephrase and reorganise the paragraph.*

*Equations*

The flux equations, currently in the results section, should be included in the data section.

*Yes, I agree that the equations should be part of Section 3.1. However, in response to the other reviewer's comment I am thinking to drop these equations.*